



# A 16-year record (2002–2017) of permafrost, active layer, and meteorological conditions at the Samoylov Island Arctic permafrost research site, Lena River Delta, northern Siberia: an opportunity to validate remote sensing data and land surface, snow, and permafrost models

Julia Boike[1,2], Jan Nitzbon[1,2,3], Katharina Anders[4], Mikhail Grigoriev[5], Dmitry Bolshi-yanov[6], Moritz Langer[1,2], Stephan Lange[1], Niko Bornemann[1], Anne Morgenstern[1], Peter Schreiber[1], Christian Wille[7], Sarah Chadburn[8,9], Isabelle Gouttevin[10], and Lars Kutzbach[11]

[1] Alfred Wegener Institute Helmholtz Center for Polar and Marine Research, Telegrafenberg A45, 14473 Potsdam, Germany

[2] Humboldt University, Geography Department, Unter den Linden 6, 10099 Berlin, Germany

[3] University of Oslo, Department of Geosciences, Sem Sælands vei 1, 0316 Oslo, Norway

[4] Heidelberg University, Department of Geography, 3D Geospatial Data Processing Research Group, Im Neuenheimer Feld 368, 69120 Heidelberg

[5] Melnikov Permafrost Institute, Siberian Branch, Russian Academy of Sciences, Merzlotnaya St., 36, Yakutsk 677010, Russia

[6] Arctic and Antarctic Research Institute, 38 Beringa Str., St. Petersburg, 199397, Russia

[7] GFZ German Research Centre for Geosciences, Telegrafenberg, 14473 Potsdam, Germany

[8] University of Leeds, School of Earth and Environment, Leeds LS2 9JT, UK

[9] University of Exeter, Department of Mathematics, Exeter EX4 4QF, UK

[10] Météo-France – CNRS, CNRM UMR 3589, Centre d'Etudes de la Neige, Grenoble, France

[11] University of Hamburg, CLISAP, Hamburg, Allende-Platz 2, 20146 Hamburg Germany

Correspondence to: Julia Boike (Julia.Boike@awi.de)

**Abstract.** Most of the world's permafrost is located in the Arctic, where its frozen organic
carbon content makes it a potentially important influence on the global climate system. The

Arctic climate appears to be changing more rapidly than the lower latitudes, but observational
data density in the region is low. Permafrost thaw and carbon release into the atmosphere is a
positive feedback mechanism that has the potential for climate warming. It is therefore partic-
ularly important to understand the links between the energy balance, which can vary rapidly
over hourly to annual time scales, and permafrost condition, which changes slowly on decadal

to centennial timescales. This requires long-term observational data such as that available from
the Samoylov research site in northern Siberia, where meteorological parameters, energy bal-
ance, and subsurface observations have been recorded since 1998. This paper presents the tem-
poral data set produced between 2002 and 2017, explaining the instrumentation, calibration,
processing and data quality control. Additional data include a high-resolution digital terrain

model (DTM) obtained from terrestrial LiDAR laser scanning. Since the data provide observa-
tions of temporally variable parameters that influence energy fluxes between permafrost, active
layer soils, and the atmosphere (such as snow depth and soil moisture content), they are suitable
for calibrating and quantifying the dynamics of permafrost as a component in earth system
models. The data also include soil properties beneath different microtopographic features (a

polygon center, a rim, a slope, and a trough), yielding much-needed information on landscape
heterogeneity for use in land surface modeling.

For the record from 1998 to 2017, the average mean annual air temperature was -12.3 °C, with
mean monthly temperature of the warmest month (July) recorded as 9.5 °C and for the coldest
month (February) -32.7 °C. The average annual rainfall was 169 mm. The depth of zero annual

amplitude niveau is at 20.8 m, and has warmed from -9.1 °C in 2006 to -7.7 °C in 2017.



The presented data are available in the supplementary material of this paper and through the

PANGAEA website (https://doi.pangaea.de/10.1594/PANGAEA.891142).



## 1 Introduction

Permafrost, which is defined as ground that remains frozen continuously for two years or more,

underlies large parts of the land surface in the northern hemisphere, amounting to about 15 million $km^2$ (Aalto et al., 2018; Brown et al., 1998; Zhang et al., 2000). The temperature range and the water and ice content of the upper soil layer of seasonally freezing and thawing ground (the active layer) determine the biological and hydrological processes that operate within this layer. Warming of permafrost over the last few decades has been reported from many circum-Arctic

boreholes (Biskaborn et al., 2018; Romanovsky et al., 2010). Warming and thawing of permafrost and an overall reduction in the area that it covers have been predicted under future climate change scenarios by the CMIP5 climate models, but at widely varying rates (Koven et al., 2012; McGuire et al., 2018). Continued observations, not only of the thermal state of permafrost but also of the multiple other types of data required to understand the changes to permafrost, are

therefore of great importance. The data required include information on conditions at the upper boundary of the soil (specifically on snow cover), on atmospheric conditions, and on various subsurface state variables (such as, e.g., soil volumetric liquid water content and soil temperature). The seasonal snow cover in Arctic permafrost regions can blanket the land surface o for many months of the year and has an important effect on the thermal regime of permafrost-

affected soils (Langer et al., 2013). The soil's water content determines not only its hydrological and thermal properties, but also the energy exchange (including latent heat conversion or release) and biogeochemical processes.

In view of these dependencies, the data sets presented here, including snow cover and the thermal state of the soil and permafrost, together with meteorological data, will be of great value

(i) for evaluating permafrost models or land surface models, (ii) for satellite calibration and

validation (cal/val) missions, (iii) in continuing baseline studies for future trend analysis (for example, of the permafrost's thermal state), and (iv) for biological or biogeochemical studies.

The Samoylov research site in the Lena River Delta of the Russian Arctic has been investigated by the Alfred Wegener Institute Helmholtz Center for Polar and Marine Research (AWI), in
collaboration with Russian and German academic partners, since 1998. The land surface characteristics and basic climate parameter data collected between 1998 and 2011 have been previously published in Boike et al. (2013). Major developments in earth system models, for example through the European PAGE21 project (www.page21.org), the Permafrost Carbon Network projects (www.permafrostcarbon.org), satellite calibration and validation missions, and obser-
vations  through the Global Terrestrial Network on Permafrost (GTN-P) have subsequently led to sustained interest from a broader modelling community in the data obtained.

In this publication we provide information on the research site and a full documentation of the data set collected between 2002 and 2017, which can be used for forcing and validation of earth system models (see e.g. Chadburn et al., 2015; Chadburn et al., 2017; Ekici et al., 2014; Ekici
et al., 2015). We present data that incorporate subsurface thermal and hydrologic components, of heat flux as well as snow cover properties, and meteorological data from the Samoylov research site, similar to the data published previously for a Spitsbergen permafrost site (Boike et al., 2018).

## 2    Site description

The Samoylov research site is located within the continuous permafrost zone on Samoylov Island in the Lena River Delta, Siberia (Figure 1). It has been a site for intensive monitoring of soil temperatures and meteorological conditions since 1998 (Boike et al., 2013).

The region is characterized by an Arctic continental climate with low mean annual air temperature of below -12 °C, very cold minimum winter air temperatures (below -45 °C), and summer air temperatures that can exceed 25 °C, a thin snow cover and a summer water balance equilibrated between precipitation input and evapotranspiration (Boike et al., 2013).

The study area of the Lena River Delta has permafrost to depths of between 400 and 600 m (Grigoriev, 1960). The active layer thawing period starts the end of May and active layer thickness reaches maximum at the end of August/beginning of September. Marked warming of this area over the last 200 years has been inferred from temperature reconstruction using deep borehole permafrost temperature measurements in the delta and the broader Laptev Sea region (Kneier et al., 2018).

Samoylov Island is located within a deltaic setting, consists of a flood plain in the western part of the island and a Holocene terrace characterized by ice-wedge polygonal tundra and larger waterbodies in the eastern part (Figure 1).

The area is generally characterized by ice-rich organic alluvial deposits, with an average ice content in the upper meter of more than 65% by volume for the Holocene terrace and of about 35% for the flood plain deposits (Zubrzycki et al., 2013). The Holocene terrace is dominated by ice wedge polygons so that a considerable volume of the upper soil layer (0–10 m) is characterized by excess ground ice (Kutzbach et al., 2004). Degradation of ice wedges, as observed throughout the Arctic (Liljedahl et al., 2016), occurs at few, localized parts of the research site (Kutzbach, 2006).

The total mapped area of the polygonal tundra on Samoylov Island (excluding the floodplain) is composed of 58% dry tundra, 17% wet tundra and 25% water surfaces, thereof 10% over-





grown water and 15% open water (Muster et al., 2012, Figure 3a). The landscape is character-
ized by polygonal tundra, i.e. a complex mosaic of low- and high-centered polygons (with moist
to dry polygonal ridges and wet depressed centers) and larger waterbodies (Muster, 2013;
Muster et al., 2012). The polygonal tundra microtopography, polygon rims, slopes, and de-
pressed centers are clearly distinguishable. Depressed polygon centers are typically water-sat-

urated or have water levels above the ground surface (shallow ponds). High-centered polygons
have inverse microtopography, i.e. drier elevated centers and wet surrounding troughs. Polyg-
onal ponds and troughs make up about 35% of the total water surface area on the island (Boike
et al., 2013).

Previous research based at the research site has focused on greenhouse gas cycling (Abnizova

et al., 2012; Knoblauch et al., 2018; Knoblauch et al., 2015; Kutzbach et al., 2004; Kutzbach et
al., 2007; Langer et al., 2015; Runkle et al., 2013; Sachs et al., 2010; Sachs et al., 2008; Wille
et al., 2008), aquatic biology (Abramova et al., 2017), upscaling of land surface characteristics
and parameters from ground-based data to remote sensing data (Cresto Aleina et al., 2013;
Muster et al., 2013; Muster et al., 2012), and hydrology (Boike et al., 2008b; Fedorova et al.,

2015; Helbig et al., 2013). Data from a few years have also been used in earth system modeling
(Chadburn et al., 2015; Chadburn et al., 2017; Ekici et al., 2014; Ekici et al., 2015) and for
modeling land surface, snow, and permafrost processes (Gouttevin et al., 2018; Langer et al.,
2016; Westermann et al., 2016; Westermann et al., 2017; Yi et al., 2014). Table 1 summarizes
the characteristics of the research site, based on data in previous publications and additional

data included in this paper.



## 3   Data description

There are three data sets presented in this paper. The first data set provides, for the first time, a full range of meteorological, thermal, and hydrologic data including a complete description and data archive of all parameters measured at the research site (Figure 2) between 2002 and 2017.

The second data set contains data, specially compiled or processed datasets for those parameters that have been measured in the period from 1998 to 2002 to obtain a long term data set, as initiated in Boike et al. (2013). The processing and level structure is described in detail in Section 4. The third data set comprises high spatial resolution data from terrestrial laser scanning of the research site completed in 2017, with resulting data sets for a digital terrain model and

for vegetation height. Additional data such as soil properties and soil carbon content are also included in this paper in order to provide a complete set of data and parameters suitable for earth system, conceptual and land surface modeling. All of these data are archived in the PANGAEA data libraries and the measuring principles and analysis are described in this paper.

Data logging between 2002 and 2013 at the research site was powered by a solar panel and a

wind turbine generator and the data was retrieved manually during site visits once or twice a year, when visual inspections were also made of the sensors. Data gaps prior to 2013 resulted mainly from problems with the site's energy supply, such as problems with the solar/wind charge controller. No other gap filling has been undertaken, but previous publications (e.g. Langer et al., 2013) suggest that reanalysis data, such as ERA-Interim, could be used for this

purpose. In Chadburn et al. (2017), a method for correcting reanalysis data to better represent the site is described and applied. Since 2013 the research site has been connected to the main electricity supply of the new Russian Research Station, resulting in much improved data collection with almost no data gaps.

Details of the sensors used are provided in the following sections, as well as descriptions of the

data quality and cleaning routine (Section 4). The instruments can be divided into above-ground

sensors (meteorological) and below-ground sensors (e.g. soil sensors). Further detailed infor-

mation on the sensors can be found in Table 2, which summarizes all of the instruments and

relevant parameters, as well as in the appendices B to H (metadata, description of instruments,

and calculations of final parameters). Figure 2 presents a time series of all parameters measured

between 2002 and 2017.

### 3.1   Meteorological station data

The standard meteorological variables described in this section were averaged over various

intervals (Table 2) with the averages, sums, and individual values all being saved hourly until

2009 and half-hourly thereafter. The sampling intervals changed as a result of different logger

and sensor setups and different available power sources. Sensors were connected directly to

data loggers. A number of different data logger models from Campbell Scientific were used

over the years (CR10X between 2002 and 2009, CR200 between 2007 and 2010, and CR1000

since 2009), together with an AM16/32A multiplexer.

### 3.1.1   Air temperature, relative humidity

Air temperature and relative humidity were measured at 0.5 m and 2 m above the ground (start-

ing with hourly averages at 2.0 m until 30 June 2009 and at 0.5 m until 26 July 2010, with half-

hourly averages thereafter) using Rotronic and Vaisala air temperature and relative humidity

probes protected by unventilated shields (Figure B1 and Table 2). According to the sensor's

manuals, the HMP45 sensors have a measurement limit of -39.2 °C, but we recorded data down

to -39.8°C. During extreme cold air temperature periods, for example, between February 1 and

March 15, 2013, stagnant air temperature values were recorded at the sensor's output limit. These data periods were manually flagged (Flag Nr. 6) using a lower temperature limit of -39.5°C.

Also of importance is the decrease in accuracy of the air temperature and humidity data with

decreasing temperature and moisture content. For example, the accuracy for the HMP45A sensor at 20 °C is ±0.2 °C, but at -40 °C it is ±0.5 °C. Campbell Scientific PT100 temperature sensors were installed on 22 August 2013 in parallel with the temperature and humidity probes, at the same heights but in separate unventilated shields, in order to circumvent this problem. Since 17 September 2017 Vaisala HMP155A air temperature and relative humidity probes were

installed which enable the full range of temperatures (below -40 °C). The uncertainty in all the temperature measurements ranges between 0.03 and 0.5 °C, depending on the sensors used; the uncertainty in the relative humidity measurements ranges between 2 and 3%. The measurement heights were not adjusted with respect to the snow surface during periods of snow cover accumulation or ablation. The lower probes (at 0.5 m) were only completely snow-covered during

two months of the 2017 winter season (16 April–11 June 2017), as observed in photographic images, and therefore this time period is flagged in the data series (Flag 8: snow-covered; Table 3).

### 3.1.2  Wind speed and direction

The wind speed and direction were measured using a propeller anemometer (R.M. Young Com-

pany 05103, Figure B2), which was aligned towards geographic north. The averaged wind direction, its standard deviation, and the wind speed were all recorded at hourly intervals until 30 June 2009 and at half-hourly intervals thereafter. The mean wind speeds and directions were



calculated using every value recorded during the measurement interval. The standard deviation of the wind direction was calculated using the algorithm provided by the Campbell Scientific

data logger.

### 3.1.3  Radiation

The net radiation was measured between 2002 and 2009 using a Kipp & Zonen NR LITE net radiometer; outgoing longwave radiation was also measured using a Kipp & Zonen CG1 pyrgeometer. Since 2009, various 4-component radiometers were used (Table 2). The averaged

values were stored at hourly intervals until 30 June 2009 and at half-hourly intervals thereafter. Further details of the measuring periods and the specifications for the different sensors can be found in Table 2. Although all radiation sensors were checked for condensation, dirt, physical damage, hoar frost, and snow coverage during the regular site visits, the instruments were largely unattended and their accuracy is therefore estimated to have been ±10%. Since August

2014 a Kipp & Zonen CNR4 four-component radiation sensor is operative, together with a CNF4 ventilation unit to prevent condensation (Figure B3). The additional heating available for the CNR4 sensor was never used.

### 3.1.4  Rainfall

Un-heated and un-shielded tipping bucket rain gauges (Environmental Measurements ARG100

and R. M. Young Company model 52203) were installed directly on the ground on 31 August 2002 (ARG100) and 26 July 2010 (52203). The Environmental Measurements ARG100 liquid precipitation probe was damaged during the winter of 2009/2010. By installing the gauge close to the ground the risk of wind-induced tipping of the bucket which would lead to false data

records, can be reduced (as observed by Boike et al., 2018). Due to the typically low snow

heights, the risk of snow coverage of the instrument is also very low.

The instruments measure only liquid precipitation (rainfall) and not winter snowfall. The tipping buckets were checked regularly during every summer by pouring a known volume of water into the bucket and carrying out frequent visual inspections for dirt or snow during each site visit.

**3.1.5  Snow depth**

The snow depth around the station has been continuously monitored since 2002 using a Campbell Scientific SR50 sonic ranging sensor (Figure B4). The sensor measures distance between the sensor and an object or surface which could be the upper surface of the snow (in winter), or the water surface, ground surface, or vegetation (in summer). On 17 July 2015 a metal plate

was placed directly beneath the ultrasonic beam to reduce the amount of noise in the reflected signal due to surface vegetation (Figure B4). The acoustic distance data obtained from the sonic sensor were temperature-corrected using the formula provided by the manufacturer (Appendix C) using the air temperature measured at the Samoylov meteorological station.

To obtain the snow depth the distance of the sensor from the surface was recorded over the

summer and the mean calculated. The recorded (corrected) winter distances are then subtracted from this mean (previous) summer value to obtain snow depth. Due to seasonal thawing the ground surface can subside by a few centimeters over the summer season (and therefore no longer be set to zero) resulting in negative heights for the ground surface level being computed. In contrast, vegetation growth and higher water levels (e.g. as observed in 2017) will result in

positive heights. The distance measurements collected during the snow-free season are not re-
moved from the series or corrected since they provide potentially useful information about these
processes.

The SR50 sensor acquires data over a discoidal surface with a radius that ranges from 0.23 m
(0.17 m$^2$) in snow-free conditions to 0.19 m (0.12 m$^2$) with 20 cm of snow. This footprint disk

is located in the center of a low-centered polygon for which the spatial variability of snow has
been investigated by Gouttevin et al. (2018). The microtopography of this polygonal tundra
(characterized by rims, slopes and polygon centers) was identified as a profound driver of spa-
tial variability in snow depth: at maximum accumulation in 2013 rims typically had 50% less
snow cover and slopes 40% more snow cover than polygon centers. However, the snow cover

within each topographical unit also exhibited spatial variability on a decimeter scale (Gouttevin
et al., 2018, Figure 4), probably resulting from underlying micro relief (notably vegetation tus-
socks) and processes such as wind erosion. This variability can affect the representativity of the
SR50-measured snow depth data and visual data obtained from time-lapse photography can
therefore be extremely important (see next section).

**3.1.6  Time lapse photography of snow cover and land surface**

In order to monitor the timing and pattern of snow melt an automated camera system (Campbell
Scientific CC640) was set up in September 2006 to photograph the land surface in the area in
which the instruments were located (Figures B5 to B7). The images are used as a secondary
check on the snow cover figures obtained from the depth sensor and are also valuable for mon-

itoring the spatial variability of snow cover across polygon microtopography. During the polar
night the image quality was found to be somewhat reduced and a second camera with a better



resolution (Campbell Scientific CC5MPX) was therefore installed in August 2015 to record high-quality images in low-light conditions over the winter period.

### 3.1.7 Atmospheric pressure

A Vaisala PTB110 sensor in a vented box was installed next to the data loggers at the meteorological station (Figure B1) in August 2014 to measure atmospheric pressure.

### 3.1.8 Water levels

The suprapermafrost ground water level, i.e. water level of the seasonally thawed active layer above the permafrost table within one polygon, was estimated using Campbell Scientific CS616

and CS625 water content reflectometer probes installed vertically in the soil and air, with the sensor's ends standing upright (Appendix D). The advantage of this method is that the sensor can remain in the soil during freezing and subzero temperatures, whereas pressure transducers need to be removed over winter and then reinstalled. For the unfrozen periods, a mixed signal is recorded from air, soil, and water.

The sensor outputs a single period measurement from which usually the bulk dielectric number and volumetric water content is calculated using an empirical polynomial calibration provided by the manufacturer We use the signal period output of the CS616 and CS625 water content reflectometer probes (Campbell Scientific, 2016) and a site-specific calibration to convert to water level with respect to the sensor base (Appendix D).



### 3.2 Subsurface data on permafrost and the active layer

#### 3.2.1 Instrument installation of the soil station and soil sampling

In order to take into account any possible effects of heterogeneity in vegetation and microto-
pography at the research site (e.g. due to the presence of polygons), instruments for measuring
the soil's thermal and hydrologic dynamics (Table 2) were installed at a number of different
positions within a low-centered polygon.

**Instrument installation and soil sampling in 2002**

A new measurement station was established in 2002, with instruments installed in four profiles
(Appendices B2 and F). Four pits were dug through the active layer and into the permafrost
(Figures B8 and B9), one at the peak of the elevated polygon rim (BS-1), one on the slope (BS-
2), a third in the depressed center (BS-3), and one above the ice wedge (Wille et al., 2003).

The surface was carefully cut and the excavated soil stockpiled separately according to depth
and soil horizon in order to be able to restore the original profile following instrument installa-
tion. The soil material is generally stratified fluviatile (and aeolian) sands and loams, with layers
of peat. The BS-1 and BS-2 soil profiles are classified as *Typic Aquiturbels* while the BS-3 soil
profile is classified as *Typic Historthel*, according to US Soil Taxonomy (Soil Survey Staff,
2010). The unfrozen topsoil layer was between 17 and 40 cm thick at the time of instrument
installation.

Sensors were installed to cover the entire depth range of the profile, i.e. from the very top,
through the active layer and into the permafrost soil. The sensors were positioned according to
the soil horizons so that every horizon in the profile was probed at least once.

Sensors were installed horizontally into the undisturbed soil profile face beneath different microtopographical features and the pits were then backfilled (Figures B10 and B13)).

Soil samples were collected before instrument installation so that physical parameters could be analyzed. Soil properties within the soil profiles, including the soil organic carbon (OC) content, nitrogen (N) content, soil textures, bulk densities, and porosities can be found in Appendix F.


The *Typic Aquiturbels* from the peak and the slope of the polygon rim show cryoturbation features due to the formation of thermal contraction polygons. The *Typic Historthel* in the polygon center, on the other hand, does not have any cryoturbation features and is characterized by peat accumulation under water-logged conditions. (Figure F1).


### 3.2.2 Soil temperature

Soil temperature sensors were installed over vertical 1D profiles in 2002 beneath a polygon center, slope, and rim. A measurement chain of temperature sensors was also installed in the ice wedge down to a depth of 220 cm. Their positions are shown in Figure B13. The temperatures were initially measured using Campbell Scientific 107 thermistors connected to a Campbell Scientific CR10X data logger with a Campbell Scientific AM416 multiplexer. Campbell Scientific's "worst case" example, with all errors considered to be additive, is given as ±0.3 °C between -25 and 50 °C. The average deviation from 0 °C determined through ice bath calibration prior to installation was 0.008 °C (maximum: 1.0 °C; minimum: -0.56 °C, standard deviation: 0.33 °C). The sensors cannot be re-calibrated once they have been installed. Phase change temperatures during spring thaw and fall refreezing are stable (the zero-curtain effect in freezing and thawing soils of periglacial regions). Assuming that freezing point depression (due to the



soil type and soil water composition) does not change significantly from year to year, these periods can be used to evaluate sensor stability. Between 2002 and 2009 the data logger and

multiplexer were not replaced which resulted in a reduced accuracy of up to ±0.7 °C during the winter freeze-back periods in 2009 for two of the sensors near to the surface (center of the polygon at -1 cm, rim of the polygon at -2 cm below ground surface, respectively).  For the remaining sensors the accuracy was better, up to ±0.5 °C. The affected data are flagged in the data series (Flag 7: decreased accuracy; Table 3). The data quality improved greatly following

the installation of a new data logger and multiplexer system (Campbell Scientific CR1000 data logger, AM16/32A multiplexer) in 2010 and the maximum offset at 0 °C during freeze-back was ±0.3 °C.

### 3.2.3  Soil dielectric number, volumetric liquid water content, and bulk electrical conductivity

Time-domain reflectometry (TDR) probes were installed horizontally in three soil profiles adjacent to the temperature probes. The fourth profile in the ice wedge records only temperature data (see Section 3.2.2., Figures B11 and B13). The TDR probes automatically record hourly measurements of bulk electrical conductivity (from 25 July 2010 only) and the dielectric number, obtained by measuring the amplitude of the electromagnetic wave over very long time

periods and the ratio of apparent probe length to real probe length (the $L_a/L$ ratio), corresponding to the square root of the dielectric number. A Campbell Scientific TDR100 reflectometer was used together with an SDMX50 coaxial multiplexers, 30 cm TDR probes (Campbell Scientific CS605) connected to a Campbell Scientific CR10X data logger between 2002 and 2010 and to a Campbell Scientific CR1000 data logger thereafter. All TDR probes were checked for

offsets following the method described in Heimovaara and de Water (1993) and in Campbell

Scientific's TDR100 manual (Campbell Scientific, 2015). The calibration delivered a probe off-
set of 0.085 (an apparent length value used to correct for the portion f the probe rods that is
covered with epoxy) which was used instead of the value of 0.09 suggested by Campbell Sci-
entific. The dielectric number $\varepsilon$ (dimensionless) and the computed volumetric liquid water val-
ues $\theta_l$ (volume/volume) in frozen and unfrozen soil are provided as part of the time series data
set. The calculation for volumetric liquid water content takes into account four phases of the
soil medium (air, water, ice, and mineral) and uses the mixing model from Roth et al. (1990)
(Appendix C).

The data are generally continuous and of high quality, and the absolute accuracy is estimated
to be better than 5%. This is estimated from the maximum deviation of calculated volumetric
liquid water content below and above the physical limits (between 0–1 or 0–100%). A probe
located at 0.37 m depth beneath the polygon rim showed a shift of about 3% (up and down) in
the volumetric liquid water content during the summers of 2009, 2013, and 2014, for which we
could not find any technical explanation. This shift is flagged in the data series (Flag 6: con-
sistency; Table 3).

Time-domain reflectometry was also used to measure the bulk soil impedance, which is related
to the soil's bulk electrical conductivity (BEC). These data were used to infer the electrical
conductivity of soil water and solute transport over a twelve-month period in the active layer
of a permafrost soil (Boike et al., 2008a). The impedance can be determined from the attenua-
tion of the electromagnetic wave traveling along the TDR probe after all multiple reflections
have ceased and the signal stabilized. The bulk conductivities were recorded hourly using the
TDR setup described above in this section. Because no calibration was done, a probe constant
($K_P$) of 1 was used for BEC waveform retrieval; Campbell Scientific suggests a $K_P$ for the

CS605 probes of 1.74. Measurements of electrical conductivity and the dielectric number were

affected by irregular spikes and possibly also by sensor drift similar to that in the soil temperature measurements. Data quality improved significantly after August 2015 when the Campbell Scientific coaxial SDM50 multiplexers were exchanged for SDM8X50 and the electrical grounding system improved. The dielectric numbers, computed volumetric liquid water contents, and soil bulk electrical conductivities can be found in the time series data set.

### 3.2.4   Ground heat flux

Two Hukseflux HFP01 heat flux plates were installed on 24 August 2002 and recorded ground heat flux at 0.18 and 0.24 m depth since then (Figure B12). The manufacturer's calibration values were used to record heat flux in W m$^{-2}$ (Hukseflux, 2016). Downward fluxes are positive and occur during spring and summer while upward heat fluxes are negative and typically occur

during fall and winter.

### 3.2.5   Permafrost temperature

The monitoring of essential climate variables (ECVs) for permafrost has been delegated to the Global Terrestrial Network on Permafrost (GTN-P) which was developed in the 1990s by the International Permafrost Association under the World Meteorological Organization. The GTN-

P has established permafrost temperature and active-layer thickness as ECV's in (1) the TSP (Thermal State of Permafrost) data set and (2) the CALM (Circumpolar Active Layer Monitoring) monitoring program (Romanovsky et al., 2010; Shiklomanov et al., 2012). A 27 m deep borehole was drilled in March 2006 with the objective to establish permafrost temperature monitoring (Figure 1, Appendix E). A 4 m long metal pipe (diameter 13 cm; extending 0.5 m above

and 3.5 m below the surface) was used for stability and to prevent the inflow of water during

summer season when the upper ground is thawed. 24 thermistors (RBR thermistor chain with an RBR XR-420 logger) were installed in August 2006, one at the ground surface and 23 between 0.75 m and 26.75 m depth, inside a PVC tube (Figure E2). A second PVC tube was inserted into the borehole and the remaining air space in the borehole was backfilled with dry

sand. Temperatures were recorded at hourly intervals, with no averaging; no data was recorded between September 2008 and April 2009. We recommend that the temperature data from the sensors at the ground surface, at 0.75, 1.75 and 2.75 m depths should not be used due to the possibility of it having been affected by the metal access pipe. The data from these sensors have not been flagged as they are of high quality, but they may not provide an accurate reflection of

the actual temperatures. They show above zero temperatures down to 1.75 m during summer in contrast to the active layer soil temperatures (Figure 2). In contrast, CALM active layer thaw never exceeded > 0.8 m since 2002 at all grid locations.

The second PVC tube was used for comparison measurements at the same depths in the borehole. The differences between the calibrated reference thermometer (PT100) showed values

between ±0.03 and ±0.33 °C (Appendix E, Table E1).

The data record shows that depth of zero annual amplitude (ZAA, where seasonal temperature changes are negligible, ≤0.1 °C) is located below 20.75 m. At 26.75 m, temperatures fluctuate with a maximum of 0.05 °C. The annual mean temperatures between the start and end of the time series, as well as minimum and maximum temperatures, are displayed in Figure 3 ("trum-

pet curve"). The permafrost warms at all depths within this 10-year period, most pronounced at the surface. At 2.75 m, the mean annual temperature increased by 5.7 °C (from -9.2 to -3.5°C), at 10.75 m by 2.8 °C (from - 9.0 to -6.2 °C) and at ZAA of 20.75 m by 1.3 °C (from -9.1 to -7.7 °C).

### 3.2.6 Active layer thaw depth

Active layer thaw depth measurements have been carried out since 2002 at 150 points over a 27.5×18 m measurement grid (Boike et al., 2013, Figure 12; Wille et al., 2003; Wille et al., 2004), by pushing a steel probe vertically into the soil to the depth at which frozen soil provides firm resistance. The data are recorded at regular time intervals, usually between June/July and the end of August, when the research site is visited. The data set shows that thawing of the

active layer continues until mid-September in some years (e.g. in 2010 and 2015). Large inter-annual variations in maximum active layer thaw depths are recorded at the end of August, ranging between a largest mean thaw depth of about 0.57 m (2011) and a smallest mean of 0.41 m (2016).

To assist in the interpretation of active layer thickness data, surface elevation change measure-

ments (subsidence measurements) have been collected since 2013 at three locations (two wet centers, one rim) using reference rods installed deep in the permafrost (Figure 1). These measurements show that a net subsidence of about 15 cm occurred between 2013 and 2017 at the rim, and smaller subsidence (-1 cm and -3 cm) at the wet centers. A net subsidence of between -1.4 to -19.4 cm between 2013 and 2017 was reported by Antonova et al. (2018) for the Yedoma

region of the Lena River Delta. Subsidence monitoring will in future be incorporated into the observational program on Samoylov Island so that active layer thaw depths can be more accurately interpreted taking into account surface changes due to subsurface excess ice melt.

## 4  Data quality control

An overview of the periods of instrumentation and parameters is provided in Figure 4.

Quality control was carried out as outlined in Boike et al. (2018) for the data set compiled from
       the Bayelva site, which is located on Spitsbergen permafrost. Quality control on observational
       data aimed to detect missing data and errors in the data, in order to provide the highest possible
       standard of accuracy. In addition to the automated processing, all data have been visually con-
       trolled and outliers manually detected, but it cannot be excluded that there are still unreasonable

values present which are not flagged accordingly. We differentiate between Level 0, Level 1,
       and Level 2 data (Table 3). Level 1 data have undergone extensive quality-control and are
       flagged with regards to equipment maintenance periods, physical plausibility, spike/constant
       value detection, and sensor drift (Table 3). Level 2 data are compiled for special purposes and
       may include combinations of data series from multiple sensors and gap-filling. Examples in this

paper of Level 2 data are soil temperature and meteorological data (air temperature, humidity,
       wind speed, and net radiation) recorded between 1998–2002 (Boike et al., 2013) that have been
       combined with data set since 2002 into a single data series, in order to obtain a long term picture
       (documentation of source data is provided in the PANGAEA data archives).

       Nine types of quality control (flags) have been used (Table 3). Data are flagged to indicate

where no data is available, or system errors, or to provide information on system maintenance
       or consistency checks based on physical limits, gradients, and plausibility.

       Due to the failure of some sensors that cannot be retrieved for repair or re-calibration (e.g.
       sensors installed in the ground), the initial accuracy and precision of the sensors may not always
       be maintained. In the case of soil temperature sensor accuracy can be estimated by analysis of

temperatures relative to the fall zero-curtain effect, assuming that the soil water composition is
       similar from year to year. Our temperature data have been checked against the fall zero-curtain
       effect and information on any reduction in accuracy is provided in Table 3 (Flag 7: decreased

accuracy). These checks are essential if subtle warming trends are to be detected and inter-
preted. The suitability of flagged data therefore depends on what it is to be used for and the
accuracy required.

The local differences between the locations from 1998 and 2002 (even though less than 50 m
meters apart), as well as differences between sensor types and accuracies, need to be considered
when interpreting longer term records. For example, relative air humidity data show marked
differences between the earlier record (1998–1999) compared to the later data set (starting in
2002). Net radiation between 1998 and 2009 showed lower values during the summer periods
compared to the summer periods between 2009 and 2017. One reason could be the change in
sensor types: during the first period, a net radiation sensor was in place, whereas during the
second period a four component radiation sensor was used.

## 5 Summary and Outlook

The climate of the period between 1998 and 2017 can be characterized as follows: The average
mean annual air temperature is -12.3 °C, with mean monthly temperature of the warmest month
(July) recorded as 9.5 °C and for the coldest month (February) as -32.7 °C. The average annual
rainfall was 169 mm and the average annual winter snow cover 0.3 m (2002–2017; no data are
available prior to 2002 for snow cover), with a maximum snow depth of 0.8 m recorded in 2017.

Since the installation in 2006, permafrost has warmed by 1.3 °C at the zero annual amplitude
niveau at 20.8 m depth. Permafrost in the Arctic has been warming and the rate of warming at
this borehole is one of the highest recorded (Biskaborn et al., 2018). Mean annual permafrost
temperatures have been increasing over the recording period at all depths, but the end-of-sea-
son's active layer thaw depth shows a marked interannual variation. Further analysis is required

to disentangle the relationships between meteorological drivers, permafrost warming, and ac-

tive layer thaw depths at this research site. The data sets described in, and distributed through,

this paper provide a basis for analyzing this relationship at one particular research site and a

means of parameterizing earth system modelling over a long observational period. The newly

collated data set will allow multi-year model validation and evaluation that includes the small-

scale microtopographic effects of permafrost-affected polygonal ground. Landscape heteroge-

neity (such as, e.g., in soil moisture) is particularly poorly represented in earth system models

and yet exerts a strong influence on the greenhouse gas balance (e.g. Kutzbach et al., 2004;

Sachs et al., 2010). As such, this data set allows the distinction between microtopographic units

(wet vs. dry) to be incorporated into modelling.  This makes this an important We will continue

to update these data sets for use in baseline studies, as well as to assist in identifying important

processes and parameters through conceptual or numerical modeling.

## 6    Data availability

The data sets presented herein are freely available as a download from PANGAEA. Permafrost

temperature and active layer thaw depth data are also available through the Global Terrestrial

Network for Permafrost (GTN-P) database (http://gtnpdatabase.org). All data are provided as

ASCII files and are freely available through the following data provider and links:

https://doi.pangaea.de/10.1594/PANGAEA.891142 with respective data sets listed in this col-

lection for time series level 1 and 2 data, terrestrial laser scanning and time lapse camera images.



## 7    Figures





**Figure 1.** Samoylov research site: **(a)** Location of Samoylov Island in the Lena River Delta, north-eastern Siberia (Landsat-7 ETM+ GeoCover 2000). **(b)** Location of instrumentation and measurement sites. **(c)** The research site under summer conditions (September 2017) and **(d)** spring conditions (April 2014; photo by T. Sachs). **(e)** Digital terrain model obtained by terrestrial laser scanning (TLS) in September 2017, and **(f)** relative heights/vegetation derived from TLS data acquired in September 2017. Further details of the methods of TLS data processing are provided in Appendix H.







**Figure 2.** Time series of Samoylov data presented in this paper. **(a)**–**(i)**: meteorological data, **(j)**–**(p)**: soil data. Seasonal average active layer thaw depth **(o)** was measured at the 150 data



points on the Samoylov CALM grid. Further details on the sensors and periods of operation are given in Table 2.

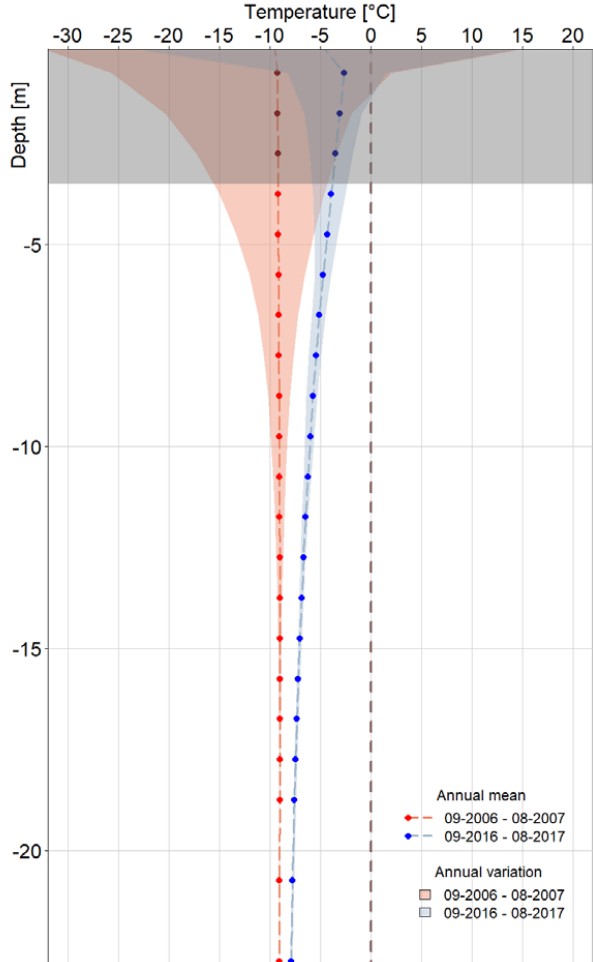

**Figure 3.** Mean annual, maximum and minimum permafrost temperatures at different depths
between 2006 and 2017, as recorded in the Samoylov Island borehole. Mean annual tempera-
tures are based on the period 1 September 2006 to 31 August 2007, and 1 September 2016 to
31 August 2017. Maximum and minimum annual variations are based on the same time period
and computed from mean daily temperatures. The upper 3.5 m below surface are shaded in
grey since we recommend not to these data for active layer thermal processes.




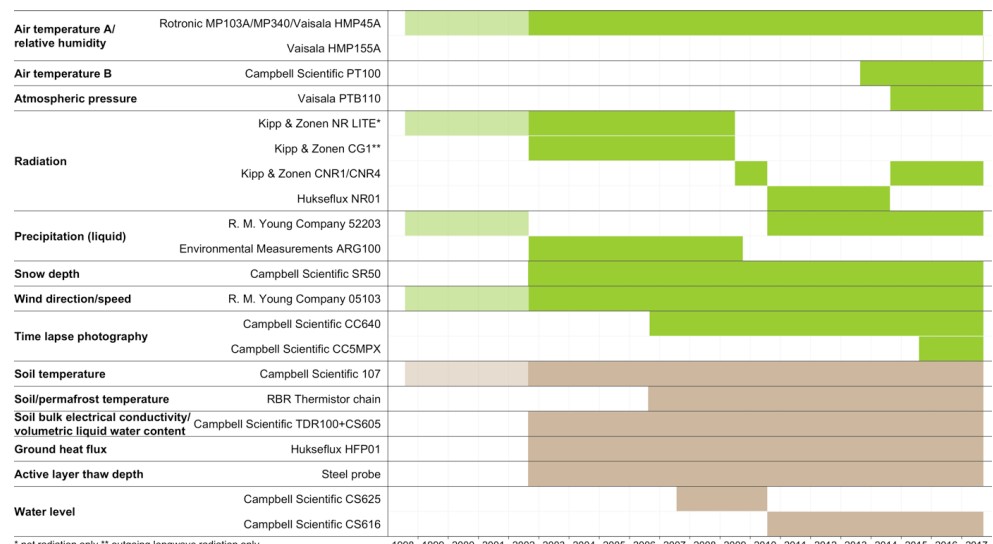

**Figure 4.** Time line for all of the parameters recorded on the Samoylov research site between 1998 and 2017. Green bars represent above-ground sensors; brown bars represent sensors installed below the ground surface. Dark brown and dark green coloring indicates a data set described in this paper (2002–2017), light brown and light green coloring indicates a previously described data set (1998–2011; Boike et al., 2013). Continuous data (light and dark colored data sets, e.g. wind speed and direction) are combined in the Level 2 product as one continuous data series for the period 1998–2017. Details of parameters for all sensors can be found in Table 2. Note that the color bars describe the sensor installation period, but data might not be available in the published data set due to sensor malfunction/failure.



## 8   Tables

**Table 1.** Site description parameters for earth system model input. Values have been computed and compiled for the Samoylov research site and surrounding areas.

| Variable | Value | Source |
|---|---|---|
| **Surface characteristics** | | |
| Summer albedo | 0.15–0.2 | Langer et al. (2011a) |
| Summer Bowen ratio | 0.35–0.50 | Langer et al. (2011a) |
| Summer roughness length (mm) | $1\times10^{-3}$ (from eddy covariance data) | Langer et al. (2011a) |
| **Snow properties** | | |
| Snow albedo | Spring period prior to melt: 0.8 (2007, 2008) | Langer et al. (2011a) |
| End of the snow ablation | 26 Apr–18 Jun (1998–2017) | Boike et al. (2013); this paper |
| Range of snow depths (end of season before ablation) (m) recorded by the SR50 sensor (thus disregarding spatial variability in snow depth) | 0.09–0.7 (1999–2017) | Boike et al. (2013) which includes two locations: 1999–2002 polygon rim; 2003–2017 polygon center |
| End of season snow density (kg m$^{-3}$) (different year and different methods) | 175–225 (field measurement) 190±10 (field measurement) 264 ±24 (based on X-ray microtomography and direct numerical simulations) | Boike et al. (2013) Langer et al. (2011b) Gouttevin et al. (2018) |
| Snow heat capacity (MJ m$^{-3}$ K$^{-1}$) | 0.39±0.02 | Langer et al. (2011b) |
| Snow thermal conductivity (W m$^{-1}$ K$^{-1}$) (bulk value for snowpack overlying vegetation/grass) | 0.22±0.03 (fitted from temperature profiles) 0.22 ±0.01 (based on X-ray microtomography and direct numerical simulations) | Langer et al. (2011b) Gouttevin et al. (2018) |
| **Soil properties** | | |
| Soil classification | Complex of *Glacic/Typic Aquiturbels* and *Histic Aquorthels* according to USDA Soil Taxonomy | Kutzbach et al. (2004) |
| Surface organic layer thickness | 0–15 cm (bare to vegetated tundra areas; up to 20 cm in wetter areas) | Boike et al. (2013) |
| Soil texture (below surface organic layer) | Sand to silt with organic peat layers of varying depths | Boike et al. (2013); Appendix F for single profiles |
| Thawed soil thermal conductivity (W m$^{-1}$ K$^{-1}$) | 0.14±0.08 (dry peat) 0.60±0.17 (wet peat) 0.72±0.08 (saturated peat) | Langer et al. (2011a) |



| Variable | Value | Source |
|---|---|---|
| Thawed soil heat capacity ($10^6$ J K$^{-1}$ m$^{-3}$) | 0.9±0.5 (dry peat) <br> 3.4±0.5 (wet peat) <br> 3.8±0.2 (saturated peat) | Langer et al. (2011a) |
| Frozen soil thermal conductivity (W m$^{-1}$ K$^{-1}$) | 0.46±0.25 (dry peat) <br> 0.95±0.23 (wet peat) <br> 1.92±0.19 (saturated peat) | Langer et al. (2011b) |
| Soil bulk density (kg m$^{-3}$) | Depth average: $0.75\times10^3$ kg m$^{-3}$ | Boike et al. (2013); Appendix F |
| Soil carbon content (g g$^{-1}$) | 0.01–0.22 | Boike et al. (2013) |
| Organic carbon stock (kg C m$^{-2}$) | 24 (for 0–100 cm) | Chadburn et al. (2017) (spatial average); Appendix F |
| Saturated hydraulic conductivity (m s$^{-1}$) | $463\times10^{-6}$ (moss layer) <br> $0.3\times10^{-6}$ (mineral layer) <br> $10.9\times10^{-6}$ <br> $130x 10^{-6}$ | Helbig et al. (2013) <br><br> Ekici et al. (2015) <br> Boike et al. (2008) |
| Clapp-Hornberger exponent (b factor) | ~4 (organic layer, typical for organic/peat) <br> ~4.5 (mineral layer, typical for sandy loam) | Beringer et al. (2001) |
| Porosity (volumetric water content at saturation) | 0.95–0.99 (organic layer) <br> 0.5–0.7 (mineral layer) | Boike et al. (2013) |
| Van Genuchten Parameters: Alpha (1 mm$^{-1}$) | sandy loam: 6 <br> peat/organic:10 | Yang and You (2013) |
| Van Genuchten Parameters: n (unit-free) | sandy loam: 1.3 <br> peat/organic: 10 | Dettmann et al. (2014) |
| **Vegetation characteristics** | | |
| Vegetation height (based on field measurements) | Wet tundra at polygon centers and on margins of polygonal ponds: moss and lichen stratum 5 cm, vascular plants stratum 30 cm. <br> Moist (dry) tundra at polygon rims and in high-center polygons: moss- and lichen stratum 5 cm, vascular plants stratum 20 cm. <br> Centers of polygonal and inter-polygonal ponds: moss stratum: 20–45 cm, vascular plants stratum 30 cm. | Knoblauch et al. (2015); Kutzbach et al. (2004); Spott (2003); this paper |
| Vegetation height (Estimates from terrestrial laser scanning) | (1) derived as mean vegetation height within a radius of 3 cm - center: mean 5.4 cm/standard deviation 2.0 cm | This paper (Appendix H) |



| Variable | Value | Source |
|---|---|---|
| | - rim: mean 4.6 cm/standard deviation 2.1 cm<br>(2) derived as maximum vegetation height (99th percentile) within a radius of 3 cm<br>- center: mean 11.7 cm/standard deviation 4.5 cm<br>- rim: mean 10.7 cm/standard deviation 5.2 cm | |
| Vegetation fractional coverage | Wet tundra at polygon centers and on margins of polygonal ponds: moss- and lichen stratum 95%, vascular plants stratum 33–55%.<br>Moist (dry) tundra at polygon rims and in high-center polygons: moss- and lichen stratum 95%, vascular plants stratum 30%.<br>Centers of polygonal and inter-polygonal ponds: moss stratum: 95%, vascular plants stratum 0–20%. | Knoblauch et al. (2015); Kutzbach et al. (2004); Spott (2003) |
| Vegetation type | Complex of G3 and W2 according to CAVM-Team (2005)<br>Moist (dry) tundra at polygon rims and in high-center polygons: Hylocomium splendens – Dryas punctata community.<br>Wet tundra at polygon centers and on margins of polygonal ponds: Drepanocladus revolvens–Meesia triquetra–Carex chordorrhiza community<br>Centers of polygonal and inter-polygonal ponds: Scorpidium scorpioides–Carex aquatilis–Arctophila fulva. | Boike et al. (2013); Knoblauch et al. (2015); Kutzbach et al. (2004); this paper |
| Max Leaf Area Index (LAI) in summer<br>(does not include moss) | 0.3 (derived from MODIS) | Chadburn et al. (2017) |
| Root depth | 30 cm (center, rim) | Kutzbach et al. (2004) |
| **Landscape** | | |
| Landscape type | Lowland polygonal tundra, mosaic of wet and moist sites | Kutzbach (2006); Kutzbach et al. (2004) |
| Bioclimate subzones | Subzone D | CAVM-Team (2005) |




**Table 2.** List of sensors, parameters, and instrument characteristics for the automated time series data from the Samoylov research site, 2002–2017. Positive heights are above the ground surface, negative heights are below the ground surface. Sensor names refer to the original manufacturer brand name (e.g. the Vaisala PTB110 air pressure sensor is distributed by Campbell Scientific as model CS106). Integration methods are average (avg), sample (spl), and sum.

| Variable | Sensor (number of sensors, if > 1) | Period of operation from | to | Height (m) | Unit | Measuring interval | Integration method | Accuracy (±) | Spectral range |
|---|---|---|---|---|---|---|---|---|---|
| **Above-ground sensors** | | | | | | | | | |
| Air temperature (A) | Vaisala HMP155A (2) | Sep 2017 | now | 0.5, 2.0 | °C | 30 s | avg 30 min | (0.226 – 0.0028×T) °C (–80 to 20 °C), (0.055 + 0.0057×T) °C (20 to 60 °C) | |
| Air temperature (A) | Rotronic MP103A/Rotronic MP340/Vaisala HMP45A (2) | Aug 2002 | Sep 2017 | 0.5, 2.0 | °C | 20 s (Aug 2002–Jul 2005),15 s (Jul 2005–Jun 2009), 10 min (Jun 2009–Jul 2009), 10 s (Jul 2009–Jul 2010), 30 s (Jul 2010–Sep2017) | avg 60 min (Aug 2002–Jun 2009), avg 30 min (Jun 2009–Jul 2009), avg 60 min (Jul 2009–Jul 2010), avg 30 min (Jul 2010–Sep 2017) | 0.5 °C (–40 to 60 °C)/0.5 °C (–40 to 60 °C)/0.2 °C (20 °C), linear increase: 0.5 °C (–40 °C), 0.4 °C (60 °C) | |
| Air temperature (B) | Campbell Scientific PT100 (2) | Aug 2013 | now | 0.5, 2.0 | °C | 30 s | avg 30 min | <0.15 °C (–100 °C), <0.1 °C (0 °C), <0.19 °C (100 °C) | |
| Relative humidity | Vaisala HMP155A (2) | Sep 2017 | now | 0.5, 2.0 | % | 30 s | avg 30 min | (1.4 + 0.032 × RH)% (–60 to –40 °C), (1.2 + 0.012 × RH)% (–40 to –20 °C), (1.0 + 0.008 × RH)% (–20 to 40 °C) | |
| Relative humidity | Rotronic MP103A/Rotronic MP340/Vaisala HMP45A (2) | Aug 2002 | Sep 2017 | 0.5, 2.0 | % | 20 s (Aug 2002–Jul 2005), 15 s (Jul 2005–Jun 2009), 10 min (Jun 2009–Jul 2009), 10 s (Jul 2009–Jul 2010), 30 s (Jul 2010–Sep2017) | avg 60 min (Aug 2002–Jun 2009), avg 30 min (Jun 2009–Jul 2009), avg 60 min (Jul 2009–Jul 2010), avg 30 min (Jul 2010–Sep 2017) | 2% (0 to 90%, 20 °C), 3% (90 to 100%, 20 °C) | |





| Variable | Sensor (number of sensors, if > 1) | Period of operation from | to | Height (m) | Unit | Measuring interval | Integration method | Accuracy (±) | Spectral range |
|---|---|---|---|---|---|---|---|---|---|
| Atmospheric pressure | Vaisala PTB110 | Aug 2014 | now | 0.7 | mbar | 30 s | avg 30 min | 1.5 mbar (–40 to +60 °C) | |
| Incoming & outgoing shortwave radiation | Kipp & Zonen CNR4 with CNF4 | Aug 2014 | now | 1.95 (Aug 2014–Jul 2016), 2.08 (Jul 2016–now) | W m$^{-2}$ | 30 s | avg 30 min | <5% (daily total, 95% confidence level) | 300–2800 nm (50% points) |
| Incoming & outgoing shortwave radiation | Hukseflux NR01 | Jul 2010 | Aug 2014 | 1.96 | W m$^{-2}$ | 30 s | avg 30 min | 10% (daily totals) | 285–3000 nm |
| Incoming & outgoing shortwave radiation | Kipp & Zonen CNR1 | Jun 2009 | Jul 2010 | 2 | W m$^{-2}$ | 10 s | avg 30 min | 10% (daily totals) | 305–2800 nm (50% points) |
| Incoming & outgoing longwave radiation | Kipp & Zonen CNR4 with CNF4 | Aug 2014 | now | 1.95 (Aug 2014–Jul 2016), 2.08 (Jul 2016–now) | W m$^{-2}$ | 30 s | avg 30 min | <10% (daily totals, 95% confidence level) | 4.5–42 μm (50% points) |
| Incoming & outgoing longwave radiation | Hukseflux NR01 | Jul 2010 | Aug 2014 | 1.96 | W m$^{-2}$ | 30 s | avg 30 min | 10% (daily totals) | 4.5–40 μm |
| Incoming & outgoing longwave radiation | Kipp & Zonen CNR1 | Jun 2009 | Jul 2010 | 2 | W m$^{-2}$ | 10 s | avg 30 min | 10% (daily totals) | 4.5–42 μm |
| Outgoing longwave radiation | Kipp & Zonen CG1 | Aug 2002 | Jun 2009 | 1.28 | W m$^{-2}$ | 20 s (Aug 2002–Jul 2005), 15 s (Jul 2005–Jun 2009) | avg 60 min | 10% (daily totals) | 4.5–42 μm (50% points) |
| Net radiation | Kipp & Zonen NR LITE | Aug 2002 | Jun 2009 | 1.35 | W m$^{-2}$ | 20 s (Aug 2002–Jul 2005), 15 s (Jul 2005–Jun 2009) | avg 60 min | 3–20% | 0.2–100 μm |
| Precipitation (liquid) | R. M. Young Company 52203 | Jul 2010 | now | 0.35 | mm | 30 s | sum 30 min | 2% (≤25 mm h$^{-1}$) | |
| Precipitation (liquid) | Environmental Measurements ARG100 | Aug 2002 | Oct 2009 | 0.4 | mm | 20 s (Aug 2002–Jul 2005), 15 s (Jul 2005–Jun 2009), 10 s (Jun 2009–Oct 209) | sum 60 min (Aug 2002–Jun 2009), sum 30 min (Jun 2009–Jul 2010) | 0.2 mm tip$^{-1}$ | |





| Variable | Sensor (number of sensors, if > 1) | Period of operation from | to | Height (m) | Unit | Measuring interval | Integration method | Accuracy (±) | Spectral range |
|---|---|---|---|---|---|---|---|---|---|
| Snow depth | Campbell Scientific SR50 | Aug 2002 | now | 1.23 (Aug 2002–Jul 2015), 1.07 (Jul 2015–now) | m | 60 min | spl 60 min | 0.4% (of distance to snow surface) | |
| Wind direction | R. M. Young Company 05103 | Aug 2002 | now | 3 | ° | 20 s (Aug 2002–Jul 2005), 15 s (Jul 2005–Jun 2009), 10 s (Jun 2009–Jul 2010), 30 s (Jul 2010–now) | avg 60 min (Aug 2002–Jun 2009), avg 30 min (Jun 2009–now) | 3° | |
| Wind speed | R. M. Young Company 05103 | Aug 2002 | now | 3 | m s$^{-1}$ | 20 s (Aug 2002–Jul 2005), 15 s (Jul 2005–Jun 2009), 10 s (Jun 2009–Jul 2010), 30 s (Jul 2010–now) | avg 60 min (Aug 2002–Jun 2009), avg 30 min (Jun 2009–now) | 0.3 m s$^{-1}$ | |
| Time lapse photography | Campbell Scientific CC640 | Sep 2006 | now | 2.2 | px | 1 day (at 12:00 local time/UTC+09) | | | |
| Time lapse photography | Campbell Scientific CC5MPX | Aug 2015 | now | 3 | px | 60 min (from 11:00 to 14:00 local time/UTC+09) | | | |
| **Below-ground sensors** | | | | | | | | | |
| Soil temperature | Campbell Scientific 107 (Aug 2002–Jul 2015: 32, Jul 2015–now: 33) | Aug 2002 | now | –0.01 to –2.71 | °C | 10 min | avg 60 min | <1.0 °C (–40 to +56 °C), <0.5 °C (–38 to +52 °C), <0.1 °C (–23 to +48 °C) | |
| Soil/permafrost temperature | RBR Thermistor chain (24) | Aug 2006 | now | 0 to –26.75 | °C | 60 min | spl 60 min | 0.1 | |
| Soil bulk electrical conductivity | Campbell Scientific TDR100 + CS605 (20) | Aug 2002 | now | –0.05 to –0.70 | S m$^{-1}$ | 60 min | spl 60 min | | |
| Soil volumetric liquid water content | Campbell Scientific TDR100 + CS605 (20) | Aug 2002 | now | –0.05 to –0.70 | % | 60 min | spl 60 min | | |
| Ground heat flux | Hukseflux HFP01 (2) | Aug 2002 | now | –0.05, –0.06 | W m$^{-2}$ | 10 min | avg 60 min | –15% to +5% (12 h total) | |
| Water level | Campbell Scientific CS616 | Jul 2010 | now | –0.115 | cm | 30 s | avg 30 min | 2.5% (≤0.5 dS m$^{-1}$, bulk density ≤1.55 g cm$^{-3}$, 0% to 50% θl) | |





| Variable | Sensor (number of sensors, if > 1) | Period of operation | | Height (m) | Unit | Measuring interval | Integration method | Accuracy (±) | Spectral range |
|---|---|---|---|---|---|---|---|---|---|
| | | from | to | | | | | | |
| Water level | Campbell Scientific CS625 | Jul 2007 | Jul 2010 | –0.15 | cm | 10 min (Jul 2007–Jul 2009), 10 s (Jul 2009–Jul 2010) | avg 60 min (Jul 2007–Jul 2009), avg 30 min (8–12 Jul 2009), avg 60 min (Jul 2009–Jul 2010) | 2.5% (≤0.5 dS m⁻¹, bulk density ≤1.55 g cm⁻³, 0% to 50% θl) | |





**Table 3.** Description of data level and quality control for data flags. Most data is flagged automatically, some are occasionally flagged manually (Flag 3: maintenance, Flag 6: plausibility). Online data transfer is not currently operational but is planned for the future.

| Flag | Meaning | Description |
|---|---|---|
| ONL | Online data | Data from online stations, daily download, used for online status check |
| RAW | Raw data | Base data from offline stations, 3-monthly backup of online data, used for maintenance check in the field |
| LV0 | Level 0 | Standardized data with equal time steps, with no gaps and in a standard data format |
| LV1 | Level 1 | Quality-controlled data including flags; quality control includes maintenance periods, physical plausibility, spike/constant value detection, sensor drifts, and snow on sensor detection |
| LV2 | Level 2 | Modified data compiled for special purposes such as combined data series from multiple sensors and gap-filled data |
| 0 | Good data | All quality tests passed |
| 1 | No data | Missing value |
| 2 | System error | System failure led to corrupted data, e.g. due to power failure, sensors being removed from their proper location, broken or damaged sensors, or the data logger saving error codes |
| 3 | Maintenance | Values influenced by the installation, calibration, and cleaning of sensors or programming of the data logger; information from field protocols of engineers |
| 4 | Physical limits | Values outside the physically possible or likely limits |
| 5 | Gradient | Values unlikely because of prolonged constant periods or high/low spikes; test within each individual series |
| 6 | Plausibility | Values unlikely in comparison with other series or for a given time of the year; flagged manually by engineers |
| 7 | Decreased accuracy | Values with reduced sensor accuracy, e.g. identified if freezing soil does not show a temperature of 0 °C |
| 8 | Snow-covered | Good data, but the sensor is snow-covered |




# Appendix A: Symbols and abbreviations

| | |
|---|---|
| $\alpha$ | geometry of the medium in relation to the orientation of the applied electrical field (Roth et al., 1990) |
| $\varepsilon_b$ | bulk dielectric number (Ka), also referred to as relative permittivity |
| $\varepsilon_l$ | temperature-dependent dielectric number of liquid water |
| $\varepsilon_i$ | dielectric number of ice |
| $\varepsilon_s$ | dielectric number of soil matrix |
| $\varepsilon_a$ | dielectric number of air |
| $\theta_l$ | volumetric liquid water content |
| $\theta_i$ | volumetric ice content |
| $\theta_s$ | volumetric soil matrix fraction |
| $\theta_a$ | volumetric air fraction |
| $\theta_{tot}$ | total volumetric water content (liquid water and ice) |
| $\bar{\rho}_{\text{bulk}}$ | average dry bulk density (kg m$^{-3}$) |
| $\Phi$ | porosity (%) |
| avg | average |
| BEC | bulk electrical conductivity (S m$^{-1}$) |
| CALM | Circumpolar Active Layer Monitoring |
| CAVM | Circumpolar Arctic Vegetation Map |
| CD$_{\text{bulk}}$ | bulk carbon density (kg m$^{-3}$) |
| ECV | essential climate variables |
| GNSS | global navigation satellite system |
| GTN-P | Global Terrestrial Network on Permafrost |
| K$_p$ | probe constant |
| $\dfrac{L_a}{L}$ | apparent length of the TDR probes (TDR data logger output) |
| MODIS | moderate resolution imaging spectroradiometer |
| N | mass fraction of nitrogen in soil (%) |





| OC | mass fraction of organic carbon in soil (%) |
| SOCC | soil organic carbon content (kg m$^{-2}$) |
| SP | signal period (ms) |
| spl | sample |
| TDR | time-domain reflectometry |
| $T_f$ | freezing temperature (°C) |
| TLS | terrestrial laser scanning |
| USDA | United States Department of Agriculture |
| WL | water level (m) |
| ZAA | zero annual amplitude |



# Appendix B: Metadata description and photos of meteorological, soil and permafrost stations and instrumentation

**B1 Meteorological station**

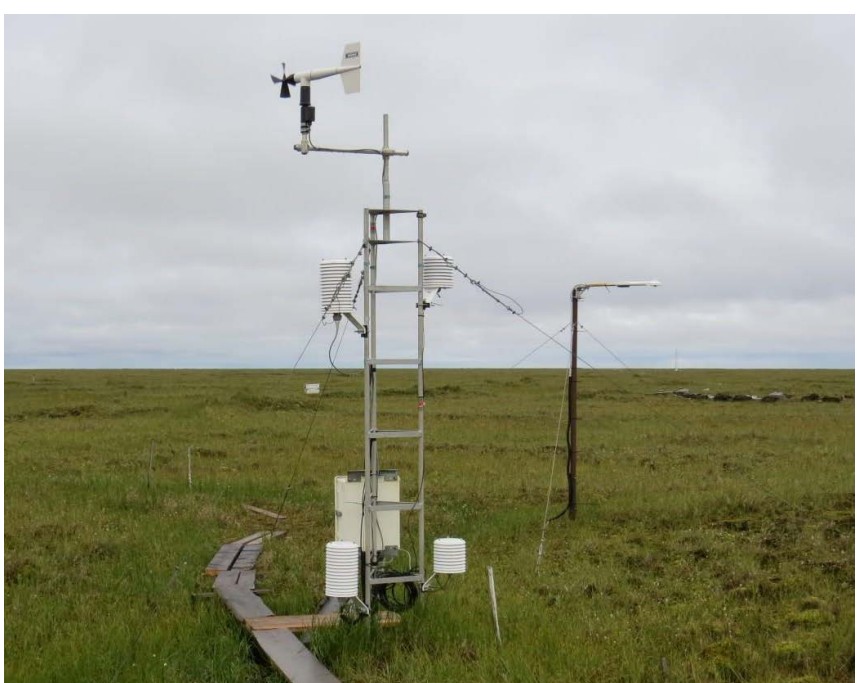

**Figure B1.** Samoylov meteorological station setup, August 2002–present (72.37001° N, 126.48106° E). Photo taken in August 2015. The two long radiation shields (left side of tower) at heights of 0.5 m and 2 m house the combined temperature and relative humidity probes (two

Vaisala HMP155A sensors were installed on 17 September 2017) and the two shorter shields (right side of tower) at the same heights contain Campbell Scientific PT100 sensors (installed on 22 August 2013) to measure air temperature only. The data logger (Campbell Scientific CR1000, installed on 30 June 2009), multiplexer (Campbell Scientific AM16/32A, installed on 27 July 2010) and barometric pressure sensor (Vaisala PTP110, installed on 22 August 2014)

are located in the white box at the back of the tower. The wind monitor and radiation sensor are shown in the figures below (Figures B2 and B3).





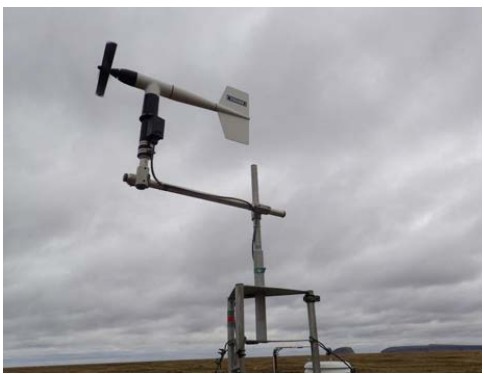

**Figure B2.** Young 05103 wind monitor for measuring wind direction and speed, installed on 31 August 2002.

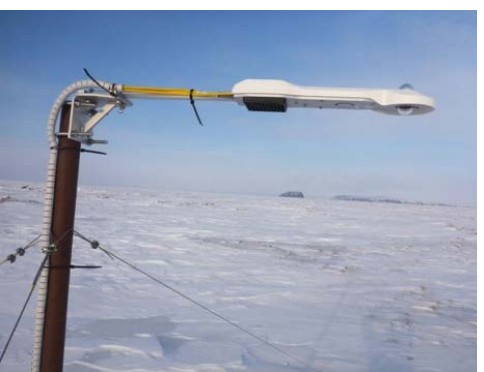

**Figure B3.** Kipp & Zonen CNR4 radiation sensor (including CNF4 ventilation unit) for measuring incoming and outgoing shortwave and longwave radiation, respectively, installed on 22 August 2014.

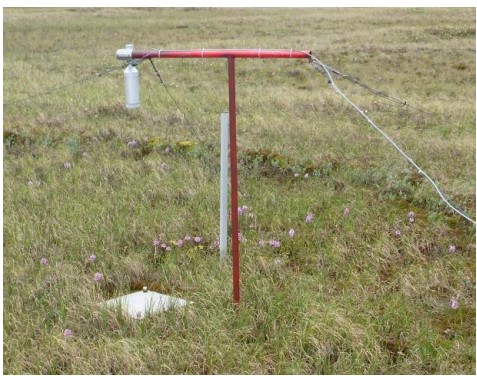

**Figure B4.** Campbell Scientific SR50 snow depth sensor, installed on 24 August 2002. An aluminum plate was installed on the ground surface beneath the sensor beam on 17 July 2016.

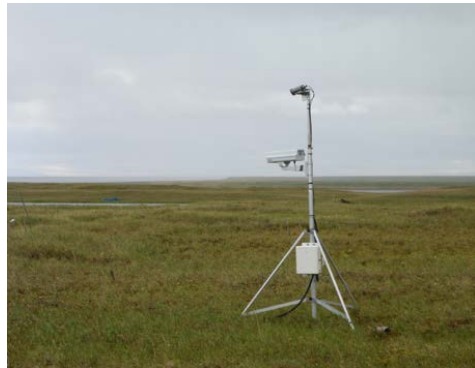

**Figure B5.** Cameras for time lapse photography of snow cover and land surface pointing towards the polygon field: a Campbell Scientific CC5MPX at the top (since 4 August 2015) and a Campbell Scientific CC640 below (since 1 September 2006). Photo taken in 2016.



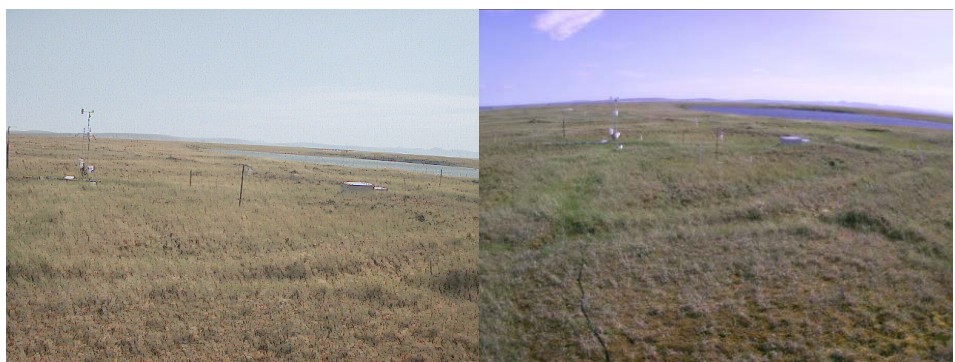

**Figure B6.** Examples of photos taken by the cameras used for time lapse photography (Figure B5) showing summer field conditions. Left photo taken by the Campbell Scientific CC640 camera (at a height of 2.2 m) and right photo taken by the Campbell Scientific CC5MPX camera (at a height of 3 m) on 7 August 2017.

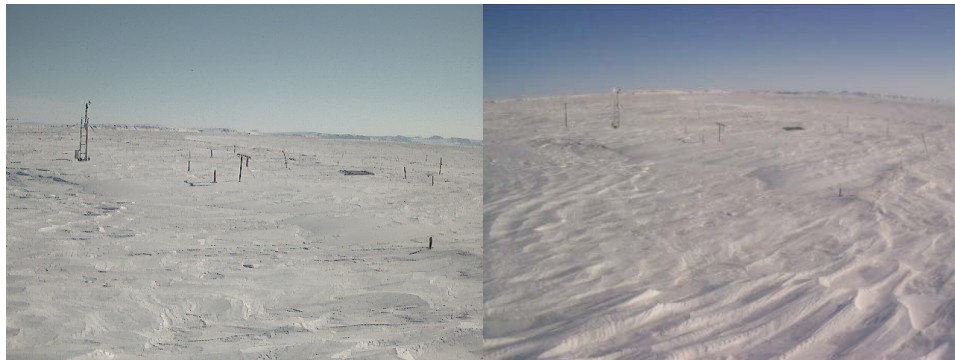

**Figure B7.** Examples of photos taken by cameras used for time lapse photography (Figure B5) showing winter field conditions. Left photo taken by the Campbell Scientific CC640 camera (at a height of 2.2 m) and right photo taken by the Campbell Scientific CC5MPX camera (at a height of 3 m) on 4 April 2017.

**B2 Soil station**



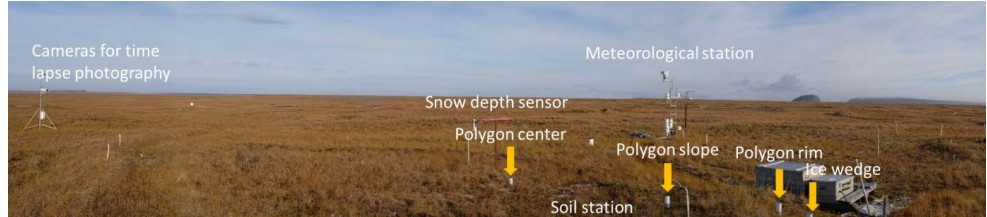

**Figure B8.** Meteorological station and soil station (consisting of sensors installed along 1D
profiles within polygon center, rim, slope, and ice wedge) with cameras for time lapse photog-
raphy pointing towards both stations for snow and surface observation.

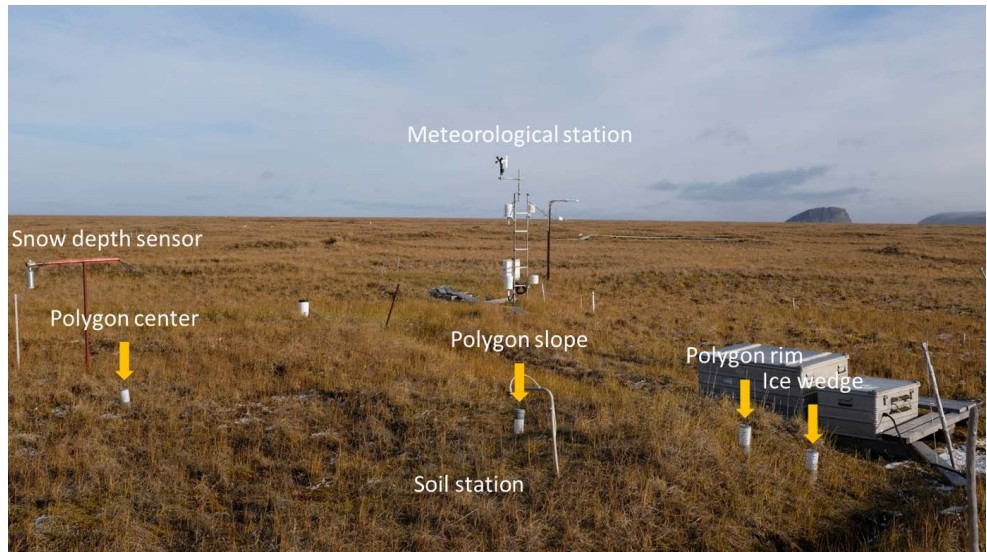

**Figure B9.** Samoylov research site in September 2017, showing locations of meteorological
station and soil station (consisting of sensors installed along 1D profiles within polygon center,
rim, slope, and ice wedge). White/grey tubes have been placed on the surface to indicate the
locations of the sub-surface sensors and their respective microtopographic locations (polygon
center, rim, slope, and ice wedge).



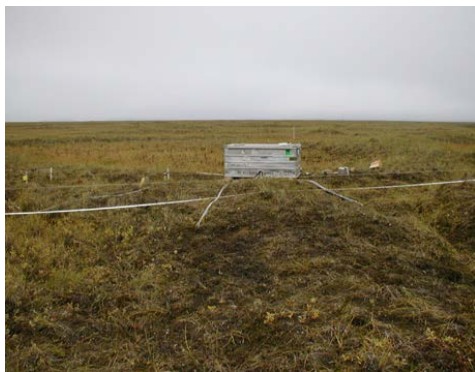

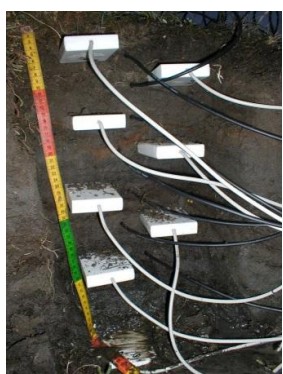

**Figure B10.** Research site after instrument installation in soil pits and subsequent refilling, August 2002. Cable strings indicate locations of center, slope, and rim profiles.

**Figure B11.** Soil volumetric liquid water content sensors: 20 Campbell Scientific CS605 TDR probes, which are connected to a Campbell Scientific TDR100 time-domain reflectometer, installed on 24 August 2002.

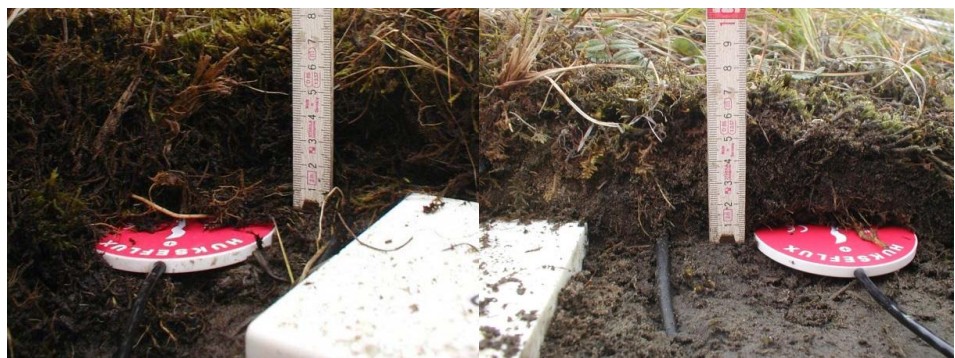

**Figure B12.** Hukseflux HFP01 ground heat-flux sensors, installed on 24 August 2002.



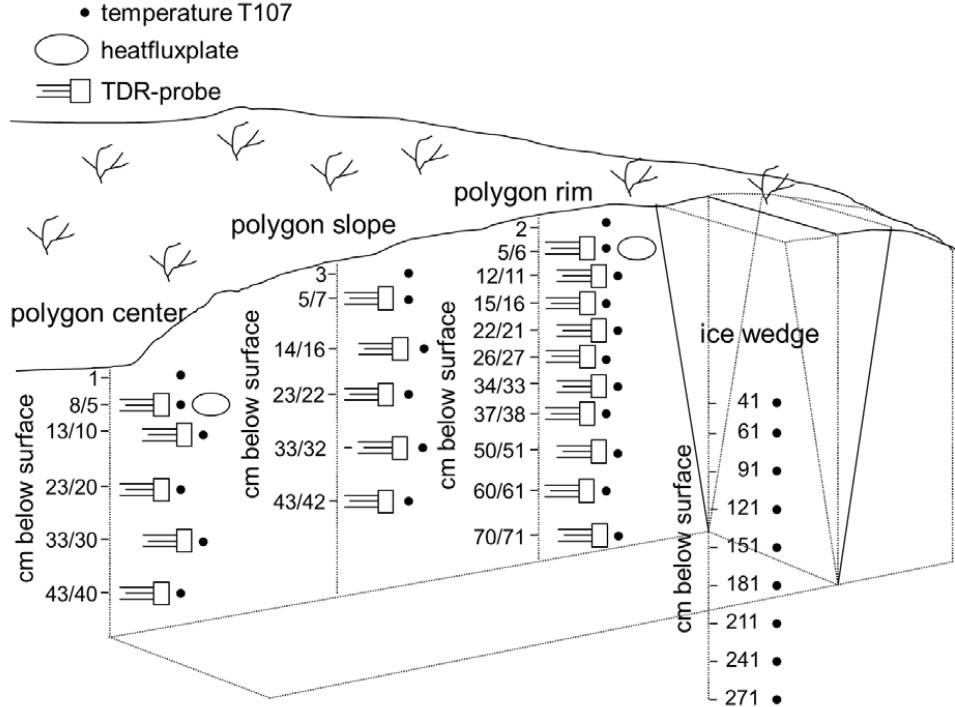

**Figure B13.** Diagram showing the sensor distribution below the polygon's center, slope, rim and inside the ice wedge, as installed on 24 August 2002. Descriptions of soil profiles and data from these profiles are provided in Appendix F.



# Appendix C: Calculation and correction of soil and meteorological parameters

### C1 Calculation of soil volumetric liquid water content using TDR

The apparent dielectric numbers were converted into liquid water content ($\theta_l$) using the semi-empirical mixing model in Roth et al. (1990). Frozen soil was treated as a four-phase porous medium composed of a solid (soil) matrix and interconnected pore spaces filled with water, ice, and air.

The TDR method measures the ratio of apparent to physical probe rod length ($\frac{L_a}{L}$) which is equal to the square root of the bulk dielectric number ($\varepsilon_b$).

The bulk dielectric number is then calculated from the volumetric fractions and the dielectric numbers of the four phases using

$$\varepsilon_b = [\theta_l \varepsilon_l^\alpha + \theta_i \varepsilon_i^\alpha + \theta_s \varepsilon_s^\alpha + \theta_a \varepsilon_a^\alpha]^{\frac{1}{\alpha}} \tag{C1}$$

A value of 0.5 was used for $\alpha$.

It is not possible to distinguish between changes in the liquid water content and changes in the ice content with only one measured parameter ($\varepsilon_b$). Equation C1 was therefore rewritten in terms of the total water content ($\theta_{tot}$) and the porosity ($\Phi$) as

$$\theta_i = \theta_{tot} - \theta_l \tag{C2}$$

Note that Equation C2 assumes the densities of liquid and frozen water to be the same, which is clearly incorrect for free phases and probably also in the pore space of soils. However, the density ratio can be absorbed into the dielectric number $\varepsilon_i$, which we do below. The resulting fluctuation of $\varepsilon_i$ is presumed to be small compared to other uncertainties.





We use

$$\theta_s = 1 - \phi \tag{C3}$$

and

$$\theta_a = \phi - \theta_l - \theta_i = \phi - \theta_{tot} \tag{C4}$$

to obtain the equation

$$\varepsilon_b = [\theta_l \varepsilon_l^\alpha + (\theta_{tot} - \theta_l)\varepsilon_i^\alpha + (1 - \phi)\varepsilon_s^\alpha + (\phi - \theta_{tot})\varepsilon_a^\alpha]^{\frac{1}{\alpha}} \tag{C5}$$

For temperatures above a threshold freezing temperature ($T > T_f$), all water is assumed to be unfrozen ($\theta_{tot} = \theta_l$). Equation C5 then reduces to:

$$\theta_l(T) = \frac{\varepsilon_b^\alpha - \varepsilon_s^\alpha + \phi(\varepsilon_s^\alpha - \varepsilon_a^\alpha)}{\varepsilon_l^\alpha - \varepsilon_a^\alpha} \quad if \ T > T_f \tag{C6}$$

For temperatures equal to or below the threshold freezing temperature ($T \leq T_f$) it was assumed that the total water content ($\theta_{tot}$) remained constant and only the ratio between volumetric liquid water content ($\theta_l$) and volumetric ice content ($\theta_i$) changed. This is a rather bold assumption as freezing can lead to high gradients of matric potential, as well as to moisture redistribution. However, since the dielectric number of ice is much smaller than the dielectric number of liquid water, the error in liquid water content measurements is still acceptable (which is not the case for ice content measurements). Under these assumptions we obtained the following equation for calculating the liquid water content of a four-phase mixture:

$$\theta_l(T \leq T_f) = \frac{\varepsilon_b^\alpha - \varepsilon_s^\alpha + \phi(\varepsilon_s^\alpha - \varepsilon_a^\alpha) + \theta_{tot}(\varepsilon_a^\alpha - \varepsilon_i^\alpha)}{\varepsilon_l^\alpha - \varepsilon_i^\alpha} \tag{C7}$$

The error of the volumetric water content measurements using TDR probes was estimated to be between 2 and 5%, which is in agreement with Boike and Roth (1997).



The availability of reliable temperature data is crucial in this approach. The liquid water content is first calculated for all times when the soil temperature was above the freezing threshold, using

Equation C5. When the soil temperature was below the freezing threshold the water content immediately prior to the onset of freezing was determined and used as the total water content ($\theta_{tot}$) for calculating the liquid water content during the frozen interval with Equation C7.

Since water in a porous medium does not necessarily freeze at 0 °C but at a temperature that depends on the soil type and water content, estimating the threshold temperature is a crucial

part of this approach. If the freezing characteristic curve is known for the material then the threshold temperature can be determined from the soil volumetric liquid water content. To avoid interpretations of frequent freezing and thawing due to soil temperature measurement errors, short-term temperature fluctuations were smoothed by calculating the mean of a moving window with an adjustable width. The smoothed temperatures were then used to trigger the switch

from one equation to the other, rather than using the original temperature time series.

The porosity values for volumetric liquid water content calculations were obtained from laboratory measurements (Appendix F) and adjusted for probe location, if necessary

**Table C1.** Porosity values for different depths and locations used for the calculation of volumetric liquid water content. Values were estimated using measured laboratory values, soil tex-

ture/horizon characteristics and TDR values at maximum saturation (=porosity).

| Depth (cm) | Location | | |
| --- | --- | --- | --- |
| | Center | Slope | Rim |
| 5 | | 93 | 67 |
| 8 | 98 | | |
| 12 | | | 67 |
| 13 | 99 | | |
| 14 | | 93 | |
| 15 | | | 67 |
| 22 | | | 67 |





| Depth (cm) | Location | | |
| --- | --- | --- | --- |
| | Center | Slope | Rim |
| 23 | 78 | 93 | |
| 26 | | | 73 |
| 33 | 99 | 100 | |
| 34 | | | 72 |
| 37 | | | 63 |
| 43 | 99 | 100 | |
| 50 | | | 60 |
| 60 | | | 55 |
| 70 | | | 64 |

## C2 Snow depth correction for air temperature

The acoustic distance sensor (Campbell Scientific SR50) measures the elapsed time between emission and return of the ultrasonic pulse. The raw distance $Dsn_{raw}$ obtained from the sensor was temperature corrected using the speed of sound at 0 °C and the air temperature at 2 m height

(Tair_200) in Kelvin (K), using the formula provided by the manufacturer (Campbell Scientific, 2007):

$$Dsn = Dsn_{raw} * \sqrt{\frac{Tair\_200 \ (K)}{273.15}} \qquad (C8)$$



## Appendix D: Metadata description and photos of installations for water level measurements

A measurement system was installed in a polygon center 3 m southeast of the meteorological station tower at 72.37001° N, 126.48106° E to allow changes in the water level to be recorded without requiring the presence of any personnel. A major disadvantage of using a common pressure transducer sensor to measure the water level is that such a device cannot withstand the long frozen arctic winter and is therefore not suitable for use when the presence of personnel is limited due to expedition schedules being restricted to summer period. A setup that can remain installed and withstand the cold winter temperatures therefore has a great advantage.

We apply vertically installed soil moisture probes to estimate water level, as described in Thomsen et al. (2000). Our sensors remained permanently in the soil with the circuit board at the base of the sensor and the parallel-connected rods pointing upwards. The base of the sensor marks the lowest measurable water level. For the water content reflectometer we measured the distance from the ground surface to the base of the sensor, where the measurement rods are connected (Figure D1), to compute water level below the ground surface. From 2007 to 2010 a Campbell Scientific CR200 data logger was connected to a Campbell Scientific CS625 probe (15 cm below the ground surface) to record the water level and two Campbell Scientific T109 sensors (1 cm and 6 cm below the surface) for temperature measurements. Since 2010 the setup has been connected to the main Campbell Scientific CR1000 logger of the meteorological station and the CS625 probe was therefore exchanged for a Campbell Scientific CS616 probe, installed 11.5 cm below the ground surface. Due to a change in data loggers in the summer of





2010, we have two setups with minor differences in the measurement probes and their installation depths, which is detailed below and visualized in Figure D1. The difference between the two water content reflectometers is the electrical output voltage, which had to be changed in order to meet the requirements of the logger. A third T109 probe was also installed 3 cm below ground surface in 2010. This setup is still in operation. These temperature data are only used to

distinguish between periods of frozen and unfrozen surface conditions. The unfrozen period, for which water levels were computed, was defined as the period for which soil temperatures at 6 cm below surface are $> 0.4°C$ during spring, and $> 0.1°C$ during fall. Below these temperatures, frozen or freezing conditions are indicated with Flag 8 (snow-covered; Table 3).

To obtain a better field calibration of the water content reflectometer a Schlumberger Mini-

Diver pressure water level sensor was installed in a well in the same polygon for 68 days of the non-frozen vegetation period in 2016. Measurements obtained from the Diver were compensated for changes in air pressure using data from the meteorological station's barometric pressure sensor (Vaisala PTB110).





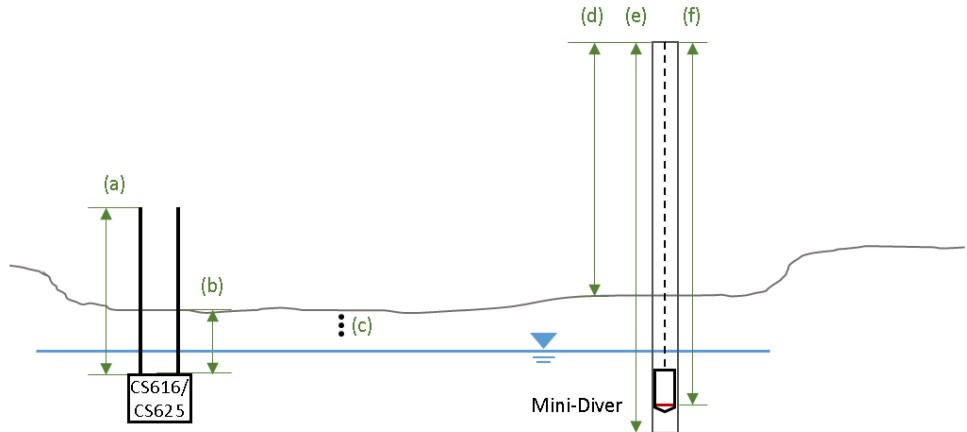

**Figure D1.** Scheme of setup of water level measurement in the polygon center: **(a)** length of the parallel measurement rods (30 cm) of the Campbell Scientific CS616/CS625 sensors, **(b)** distance from the sensor base to the ground surface (CS625: -15 cm; CS616 -11.5 cm), **(c)** Campbell Scientific T109 probes at depths of -1 cm, -3 cm, -6 cm below surface, **(d)** height of the well above the ground surface (45.5 cm), **(e)** length of the well (70 cm), **(f)** distance from the top of the well to the water pressure measurement level of the Schlumberger Mini-Diver. The difference between the ground level at CS616 and Mini-Diver locations is 3 cm. Blue line illustrates water level, green line the ground surface.



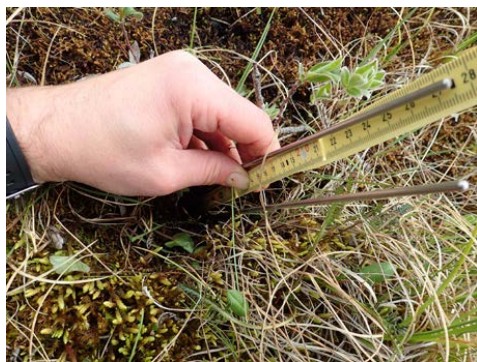
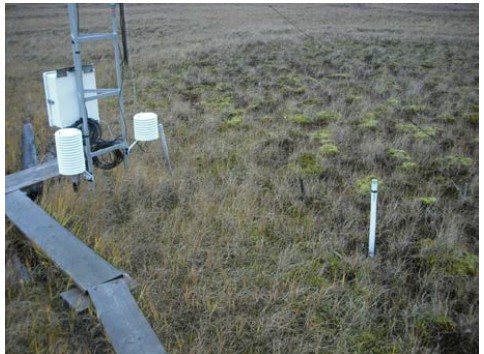

**Figure D2.** Campbell Scientific CS616 vertical probe installation. Water level measurement is done manually with a ruler.

**Figure D3.** Installation for water level measurements using a permeable ground water measurement tube with a Schlumberger Mini-Diver.

### D1 Calculation and correction of water level measurements

The measured output period (SP) from the Campbell Scientific CS616 or CS625 probes were

converted into the height of the water level above the sensor base (WL) using two polynomial

functions derived from an empirical field experiment to determine the correlation between results from the CS616/CS625 probes and those from a Mini-Diver.

The two regressions represent different water level regimes (low and higher water levels) recorded by the CS616/CS625 sensor. The results of this experiment showed a low accuracy for

very low water levels (1.5 cm or less above the sensor base) resulting in output periods of SP < 19 µs, which were excluded from the data series. For values > 19 µs, the following formulas are applied to obtain WL data from the CS616 and CS625 probe output:





$$WL = 0.01831394\ SP^3 - 1.2398\ SP^2 + 28.84699187\ SP \\ - 224.41499308 \tag{D1}$$

for SP < 27 µm and


$$WL = 0.06194726\ SP^2 - 1.7673294\ SP + 13.66709591 \tag{D2}$$

for SP > 27 µm.

The mean deviation of the calculated WL values from the values measured with the Diver was
(0.034 cm) with a standard deviation of 0.29 cm (number of values: 2679).

Note that WL is given relative to the ground surface in the time series data and reported in

meters. To obtain water level relative to the ground surface (WL$_{gs}$) from Level 1 data, the fol-

lowing calculation is suggested:

for CS616                    $WLgs = WL - 0.115$ m                    (D3)

for CS625                    $WLgs = WL - 0.15$ m                    (D4)

Special post-processing of the CS625 sensor readings was carried out from between 06 July

2009 to 26 July 2010, as no probe output periods were logged over this period. Instead, volu-

metric liquid water content ($\theta_l$,) was stored on the CR200 logger, calculated from the CS625

probe output and using a formula from the sensor's manual (Campbell Scientific, 2016). The $\theta_l$,

values were converted to SP values using formula D5:

$$SP = 39.12153154\ \theta_l^3 - 61.59657836\ \theta_l^2 + 56.7054971\ \theta_l \\ + 15.37001712 \tag{D5}$$

We compared the calculated WL with manual distance measurements taken in the field over

the years (n = 12). The largest differences between TDR derived and manual measurements



was 2 cm. This includes all measurement errors, such as sensor movement (probes are not anchored into the permafrost, they can potentially move with the seasonal heaving, subsiding of the active layer), difficulties in defining the ground surface (which is covered by mosses and grasses).



## Appendix E: Metadata description and photos of the borehole, 2006

A borehole was drilled at 72.36941° N, 126.47612° E into the permafrost during the spring of 2006. Drilling started with 146 mm diameter down to 4 m depth and continued with 132 mm diameter down to 26.75 m depth. A 4 m long metal stand pipe (diameter 13 cm) was used for stability and to prevent the inflow of water during summer season into the borehole. The metal pipe extends 0.5 m above and 3.5 m below the ground surface (Figures E1 and E2).

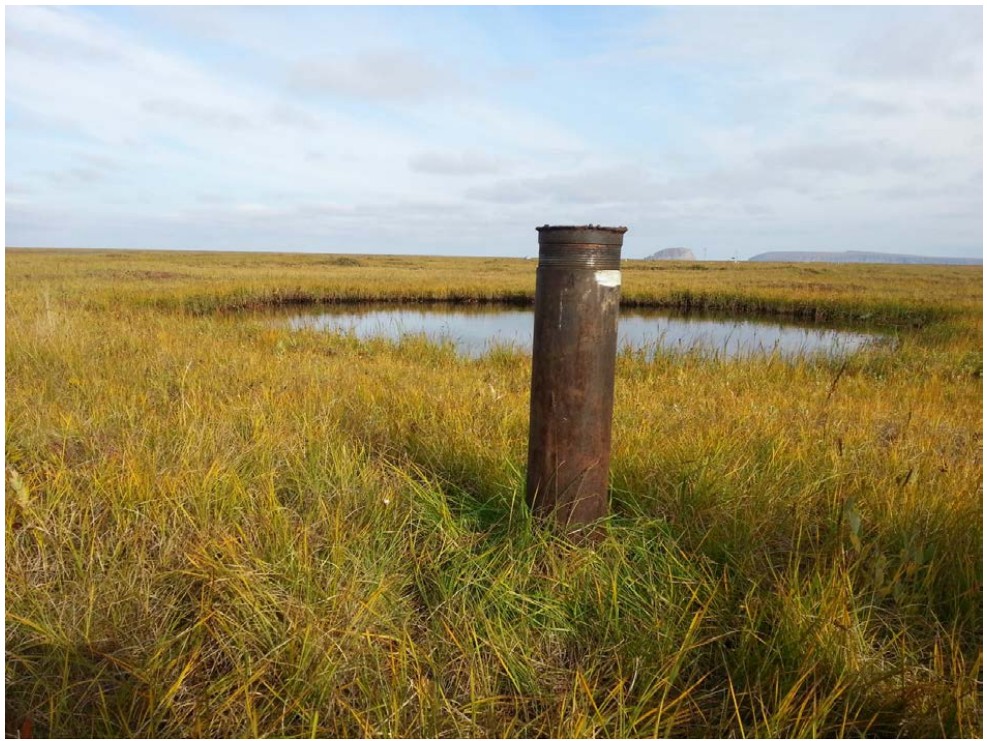

**Figure E1.** Location of the 2006 borehole showing the proximity of the borehole to a small lake. The metal pipe extends 0.5 m above the ground surface.





A thermistor chain with 24 temperature sensors (RBR thermistor chain with an XR-420 logger)

was inserted into a close fitting PVC tube (4 cm inside, 5 cm outside diameter) and installed in

the borehole on 21 August 2006, down to a depth of 26.75 m (Figure E2).

A second PVC tube with the same dimensions as the first tube was also inserted into the bore-

hole to permit additional (geophysical and calibration) measurements to be made in the future.

The remaining air space in the borehole was backfilled with dry sand. The outside metal pipe

(used for drilling and to prevent inflow of water) which stands 0.5 m above ground surface, was

closed at the top and was covered with a wooden shield, which was renewed in 2015.

The accuracy of the temperature sensors of the thermistor chain is reported by RBR to be

±0.005 °C between -5 °C and 35 °C. However, direct comparison with a high precision refer-

ence PT100 temperature sensor (certified to be accurate to ±0.01 °C between -20 and 30 °C) at

six different depths in the borehole between 9 and 17 August 2014 showed the accuracy of the

RBR XR-40 temperature sensors to be approximately ±0.03 °C at depths ≥8.75 m (Table E1).

The deviation increased with decreasing depth, e.g. between -7.75 m and -1.75 m the deviation

was ±0.33 °C and at -0.75 m it was ±0.65 °C. This increase in deviation towards the surface

may be because (a) the chain was installed in sand whereas the calibration thermometer was in

air and could therefore possibly have been affected by air circulation, or (b) the temperature

gradient becomes steeper with decreasing height below the surface and thus small differences

between the measuring heights of the two sensors will have a larger impact on temperatures as

the surface is approached. The offset of the reference thermometer at exactly 0 °C was 0.01 °C,

and the average statistical accuracy ($U_{k=2}$) is given by the manufacturer as 0.1083 °C. During



calibration in the borehole the temperature was given time to stabilize (i.e. until the recorded temperature change was less than ±0.03 °C) before being recorded (Table E1).

Continuous measurements have been obtained since mid-August 2006 from sensor depths of 0.00, 0.75, 1.75, 2.75, 3.75, 4.75, 5.75, 6.75, 7.75, 8.75, 9.75, 10.75, 11.75, 12.75, 13.75, 14.75, 755  15.75, 16.75, 17.75, 18.75, 20.75, 22.75, 24.75, and 26.75 m below the ground surface. We recommend that the temperature data from the three sensors at 0, 0.75, 1.75 and 2.75 m depths, a should not be used due to the possibility of it having been affected by the metal access pipe. The data from these sensors have not been flagged as they are of high quality, but they may not provide an accurate reflection of the actual soil temperatures.

Construction of the new Russian Samoylov Island Research Station started in September 2011 and was completed in summer 2012. The new research station included a water supply from a nearby lake. The water supply system (Figures E3 and E4) is an above-ground structure that is likely to affect the wind and hence the accumulation of snow on the tundra surface. Visual inspection in the vicinity of the borehole in April 2016 suggested an increased snow accumu-
lation around this location since construction of the water supply system. A new borehole was drilled in April 2018 down to 61 m, far away from the research station and associated structures. A new temperature chain will be installed in the fall of 2018 to provide deeper permafrost data, as well as observations from a second borehole.




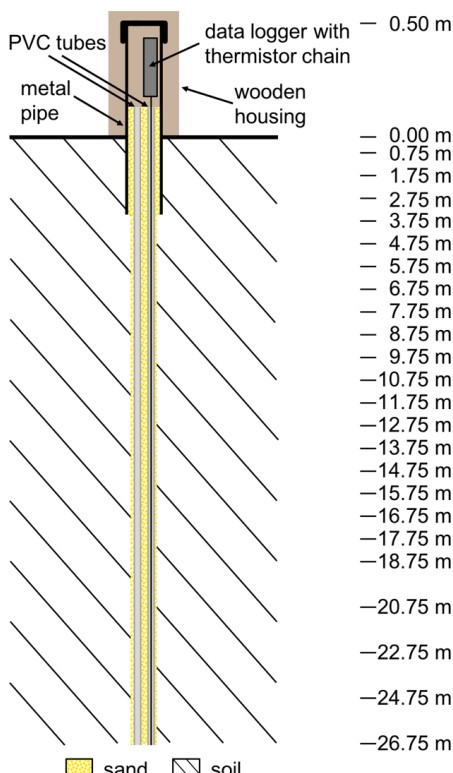

**Figure E2.** Thermistor setup showing their depths within the borehole, as installed in 2006.
The metal pipe extends 0.5 m above ground surface. Note the differences in scale between
above and below ground surface.



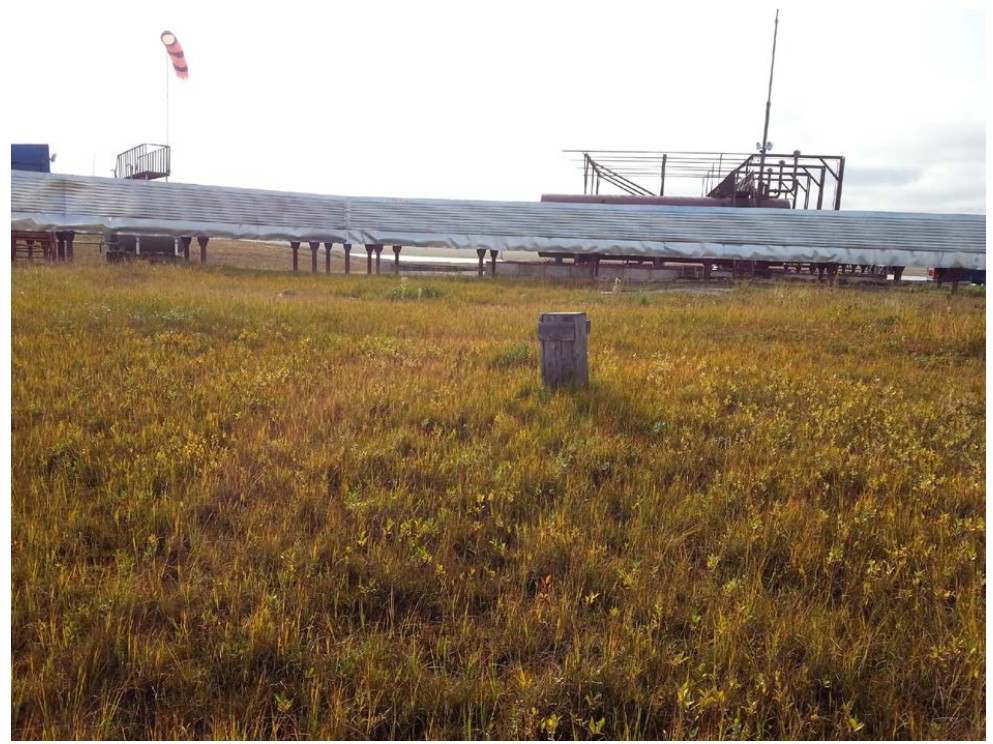

**Figure E3.** Location of the 2006 borehole (wooden box) showing the proximity of the new

water supply system (since 2013: silver metal structure extending above the tundra surface).




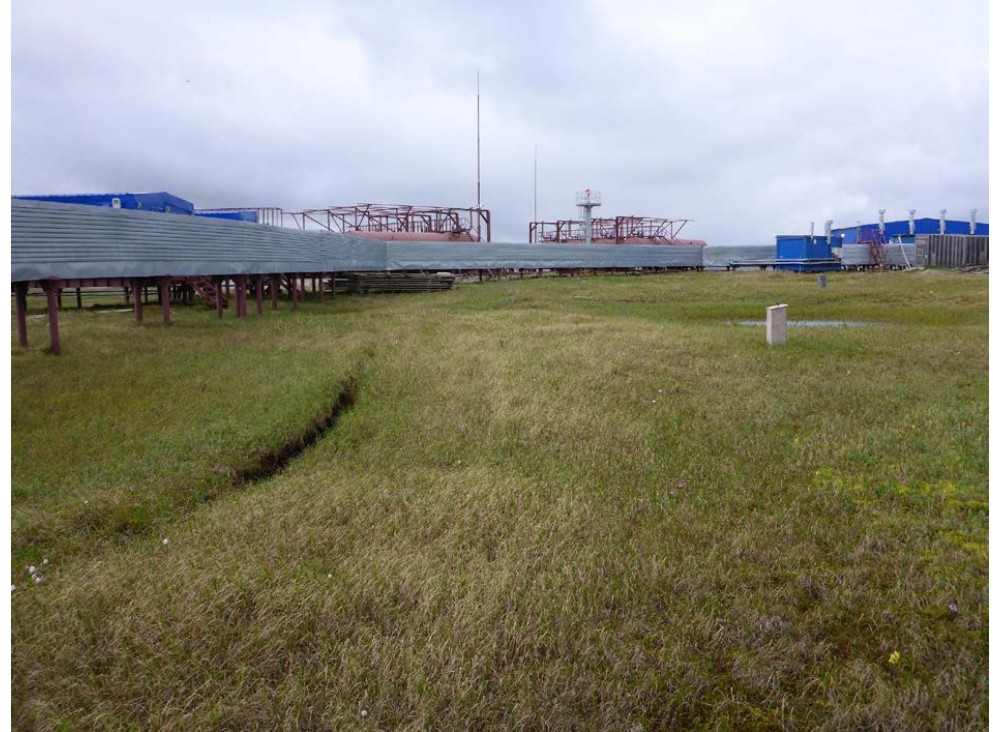

**Figure E4.** Borehole location (wooden box on right side) with the new Samoylov research station and water supply system (silver metal structure extending above the tundra surface).





**Table E1.** In situ calibration of thermometers in borehole between 9 and 17 August 2014. Comparison measurements were made in the 27 m borehole using a certified PT100 thermometer (Service für Messtechnik Geraberg DTM 3000).

| Depth | RBR Thermistor chain | DTM 3000 | Standard deviation |
|---|---|---|---|
| (m) | (°C) | (°C) | (°C) |
| 0.00 | 8.88 | 8.78 | 0.10 |
| -0.75 | 0.98 | 1.63 | -0.65 |
| -1.75 | -1.95 | -1.68 | -0.27 |
| -2.75 | -3.52 | -3.22 | -0.30 |
| -3.75 | -4.79 | -4.46 | -0.33 |
| -4.75 | -5.75 | -5.46 | -0.29 |
| -5.75 | -6.46 | -6.23 | -0.23 |
| -6.75 | -6.91 | -6.74 | -0.17 |
| -7.75 | -7.15 | -7.05 | -0.10 |
| -8.75 | -7.30 | -7.24 | -0.06 |
| -9.75 | -7.37 | -7.35 | -0.03 |
| -10.75 | -7.43 | -7.42 | -0.01 |
| -11.75 | -7.50 | -7.50 | 0.00 |
| -12.75 | -7.57 | -7.57 | 0.00 |
| -13.75 | -7.64 | -7.64 | 0.00 |
| -14.75 | -7.73 | -7.74 | 0.00 |
| -15.75 | -7.82 | -7.82 | 0.00 |
| -16.75 | -7.90 | -7.89 | -0.01 |
| -17.75 | -7.99 | -7.98 | -0.01 |
| -18.75 | -8.06 | -8.06 | 0.00 |
| -20.75 | -8.20 | -8.18 | -0.02 |
| -22.75 | -8.30 | -8.27 | -0.03 |
| -24.75 | -8.35 | -8.36 | 0.01 |
| -26.75 | -8.41 | -8.41 | 0.00 |




# Appendix F: Data from soil profiles

**Table F1.** Soil data from the BS-1 (polygon rim) and BS-3 (polygon center) soil pits, which were sampled and had instruments installed in 2002. The location of the soil profiles is described in Wille et al. (2003) and shown in Figure B8 and B9. Photos of the soil profiles can be seen in Figure F1 below. Grain size classification is according to Folk (1954) where S = sand, s = sandy, Z = silt, z = silty, M = mud, m = muddy, C = clay, and c = clayey. Other abbreviations

used are OC for the mass fraction of organic carbon in soil, N for the mass fraction of nitrogen in soil, $CD_{bulk}$ for the organic carbon density in bulk soil, and SOCC for the soil organic carbon content for each soil horizon. The soil horizon designations are according to the USDA's Soil Survey Staff (2010). The BS-1 and BS-2 soil profiles are classified as *Typic Aquiturbel* and the BS-3 soil profile is classified as *Typic Historthel* according to the US Soil Taxonomy (Soil

Survey Staff, 2010). Analysis of the physical properties of soil was done according to DIN 19683 (1973). OC and N were determined following removal of inorganic carbon and dry combustion at 900 °C (DIN ISO 10694).

| Sample-ID | Date | Profile number | Horizon | Depth below surface | Horizon thickness | OC | N | Dry bulk density | Average dry bulk density | $CD_{bulk}$ | SOCC | Φ | Grain size class |
|---|---|---|---|---|---|---|---|---|---|---|---|---|---|
| | | | | (cm) | (cm) | (%) | (%) | (g cm⁻³) | (g cm⁻³) | (kg m⁻³) | (kg m⁻²) | (%) | |
| LD02-6983 | 2002 | BS-1 | Oi | 0 | 3 | 16.9 | 0.34 | | 0.25 | 42 | 1.3 | n.d. | organic |
| LD02-6984 | 2002 | BS-1 | Ajj1 | 3 | 4 | 3.1 | 0.17 | 0.96 | 0.96 | 29 | 1.2 | 64 | zS |
| LD02-6985 | 2002 | BS-1 | Ajj2 | 7 | 8 | 1.8 | 0.11 | 1.25 | 1.25 | 22 | 1.8 | 52 | zS |
| LD02-6986 | 2002 | BS-1 | Bjjg1 | 15 | 8 | 2.4 | 0.15 | 1.25 | 1.25 | 30 | 2.4 | 54 | zS |
| LD02-6987 | 2002 | BS-1 | Bjjg2 | 23 | 6 | 5.6 | 0.31 | 0.88 | 0.88 | 49 | 2.9 | 68 | sZ |
| LD02-6988 | 2002 | BS-1 | Bjjg3 | 29 | 5 | 3.3 | 0.17 | 0.95 | 0.95 | 31 | 1.5 | 65 | S |
| LD02-6989 | 2002 | BS-1 | Bjjg4 | 34 | 6 | 5.5 | 0.27 | 0.86 | 0.86 | 48 | 2.9 | 67 | sZ |
| LD02-6990 | 2002 | BS-1 | Bjjgf1 | 40 | 15 | 2.5 | 0.17 | | 0.9 | 23 | 3.4 | n.d. | sZ |



| Sample-ID | Date | Profile number | Horizon | Depth below surface | Horizon thickness | OC | N | Dry bulk density | Average dry bulk density | $CD_{bulk}$ | SOCC | Φ | Grain size class |
|---|---|---|---|---|---|---|---|---|---|---|---|---|---|
| | | | | (cm) | (cm) | (%) | (%) | (g cm$^{-3}$) | (g cm$^{-3}$) | (kg m$^{-3}$) | (kg m$^{-2}$) | (%) | |
| LD02-6991 | 2002 | BS-1 | Bjjgf2 | 55 | 10 | 1.7 | 0.13 | | 0.9 | 15 | 1.5 | n.d. | sZ |
| LD02-6992 | 2002 | BS-1 | Bjjgf3 | 65 | 35 | 1.4 | 0.11 | | 0.9 | 13 | 4.5 | n.d. | sZ |
| LD02-7007 | 2002 | BS-3 | Oi | 0 | 15 | 19.5 | 0.62 | 0.1 | 0.1 | 20 | 2.9 | 99 | organic |
| LD02-7008 | 2002 | BS-3 | A | 15 | 19 | 5.1 | 0.14 | 0.65 | 0.65 | 33 | 6.3 | 78 | S |
| LD02-7009 | 2002 | BS-3 | Bgf | 34 | 16 | 5.2 | 0.22 | | 0.9 | 46 | 7.4 | n.d. | S |

The organic carbon density in bulk soil $CD_{bulk}$ (kg m$^{-3}$) was calculated using the mass fraction of organic carbon in soil OC, the average dry bulk density $\bar{\rho}_{bulk}$, and the following formula:

$$CD_{bulk} = OC * \bar{\rho}_{bulk} \tag{F1}$$

The organic carbon content for each soil horizon SOCC (kg m$^{-2}$) was calculated using the mass fraction of organic carbon in soil OC, the average dry bulk density $\bar{\rho}_{bulk}$, the horizon thickness, and the following formula:

$$SOCC = OC * \bar{\rho}_{bulk} * \text{horizon thickness} = CD_{bulk} * \text{horizon thickness} \tag{F2}$$




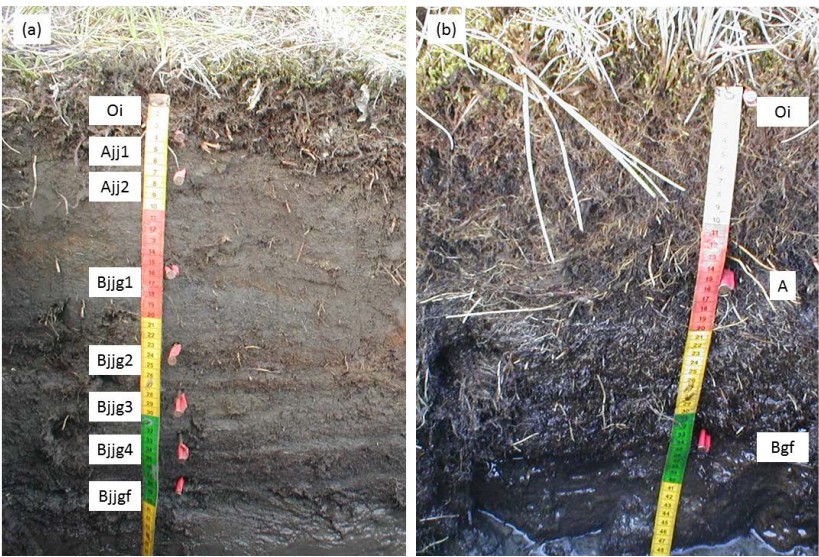


**Figure F1.** Photographs of soil profiles **(a)** BS-1 (*Typic Aquiturbel*) at the peak of a polygon rim and **(b)** BS-3 (*Typic Historthel*) at a polygon center. Designations of soil horizons according to US Soil Taxonomy (Soil Survey Staff, 2010). Horizon labels are positioned at the upper boundary of the respective horizon.



## Appendix G: names of the variables and units for data files

**Table G1.** Overview of all variables published as time series. Some variables have _center, _rim and _slope as location index (Section 3.2). Additional Level 2 data is published for the variables air temperature, relative humidity, precipitation, wind speed and direction, net radiation, soil temperatures, and soil volumetric liquid water content, which is indicated by "_lv2" in the column name. If an air temperature sensor is covered by snow and thus measures snow temperature, this is indicated by a Flag 8 in the data.

| Variable | Column name | Units |
|---|---|---|
| air/snow temperature | Tair_(height in cm) | °C |
| relative humidity | RH_(height in cm) | % |
| atmospheric pressure | PA | kPa |
| incoming shortwave radiation | SwIn | W m$^{-2}$ |
| outgoing shortwave radiation | SwOut | W m$^{-2}$ |
| incoming longwave radiation | LwIn | W m$^{-2}$ |
| outgoing longwave radiation | LwOut | W m$^{-2}$ |
| net radiation | RadNet | W m$^{-2}$ |
| wind speed | Vwind_(height in cm) | m s$^{-1}$ |
| wind direction | Dirwind_(height in cm) | ° |
| wind direction standard deviation | Dirwind_sd_(height in cm) | ° |
| active layer thaw depth | Dal_(ID) | cm |
| soil/permafrost temperature | Ts_(depth in cm) | °C |
| soil bulk electrical conductivity | Cond_(depth in cm) | S m$^{-1}$ |
| soil dielectric number | E2_(depth in cm) | – |
| soil volumetric liquid water content | Vwc_(depth in cm) | – |
| ground heat flux | G | W m$^{-2}$ |
| precipitation (liquid) | Prec | mm |
| snow depth | Dsn | m |
| water level | WL | m |



# Appendix H: Terrestrial laser scanning – analysis of 2017 data


3D point cloud data was acquired for several polygons around the meteorological, soil and CALM sites by terrestrial laser scanning (TLS) on 12 September 2017, using a RIEGL VZ-400 3D TLS instrument. According to the manufacturer's specifications the TLS instrument measures 3D coordinates with an accuracy of 5 mm and a precision of 3 mm (RIEGL LMS,

2017). We captured the full extent of the research site, which has dimensions of approximately 70×70 m, from ten scan positions with a horizontal and vertical point spacing of 3 mm at 10 m measurement range. The single point clouds were registered into a common coordinate system using five cylindrical reflectors placed around the research site during the TLS data acquisition so that they were visible from all scan positions. Mean residual distances per scan position

between the cylindrical reflectors amounted to 1.6 cm, with a standard deviation of 0.8 cm.

The registered 3D point cloud data set was georeferenced using high-accuracy global positioning measurements recorded with a global navigation satellite system (GNSS). We obtained GNSS measurements in static phase observation mode with a Leica Viva GS10 as the base station receiver and a GS15 mobile rover unit (Leica Geosystems, 2012a, b). According to the

manufacturer's specifications (Leica Geosystems, 2012a) this mode achieves a measurement accuracy of 3 mm horizontally and 3.5 mm vertically with respect to the local reference frame established by the base station. The scan positions were georeferenced and registered using the RiSCAN PRO software (version 2.1.1, RIEGL LMS, 2016).



The raw data set was filtered using a statistical outlier removal (SOR, Rusu and Cousins, 2011)

to remove spatially isolated points as outliers from the point cloud, with the number of neighbors set to 10 and the standard deviation multiplier threshold to 1.0.

A digital terrain model (DTM) representing the ground surface elevation was derived from this pre-processed data set. To determine the ground surface elevation the 3D TLS points were first classified into ground and non-ground points. For this we used a minimum approach, classify-

ing all points within a search radius of 5.0 cm that were at less than 0.05 cm vertical distance from the minimum point elevation as ground points. This vertical distance threshold is included to take into account position uncertainties of the TLS acquisition. The ground points in the 3D TLS data set are subsequently rasterized into the final DTM (with a cell size of 5.0 cm) using a robust moving planes interpolation strategy (TU Wien, 2016).

For evaluation purposes the DTM was compared to 27 GNSS measurements of the ground surface that were obtained during the TLS data acquisition. The data sets were compared by taking the difference between GNSS-based elevation measurements and the corresponding DTM pixel values. Statistical analysis of these differences in ground surface elevation yielded a mean difference of 3.7 cm, a median difference of 1.7 cm, and a standard deviation of 5.1 cm.

Differences were mainly within the accuracy ranges of TLS point cloud registration and GNSS positioning. Larger positive differences (> 2.0 cm) indicated an overestimation of ground surface elevation in the TLS point cloud. Where dense, short vegetation is present an error is introduced to the estimated ground surface elevation as the laser beam does not hit the ground surface at every local area in the site. This is to be expected, particularly for larger distances

from the scan positions as the incidence angle from the TLS instrument has a direct effect on

the penetration depth of the laser beam (Marx et al., 2017).

Relative height above the ground surface was derived as vertical distance of TLS points to the

ground surface. The DTM was used to calculate the vertical distance to the ground surface for

every 3D point in the TLS point cloud. A raster of relative height values was generated using

the 99th percentile of the relative height attribute per raster cell, with a cell size of 5 cm. Fur-

thermore, a raster of mean relative heights above ground surface was generated that could pro-

vide an estimate of the vegetation height and volume within each 5 cm raster cell. With regard

to the vegetation height values derived from the TLS data, it should be noted that the heights

could be underestimated when compared to actual field measurements, for which there are two

possible explanations. Firstly, overestimation of the ground surface elevation (where the laser

beam does not fully penetrate the vegetation) reduces the calculated relative (vegetation) height.

Secondly, the sampling of the laser scanning process with the given 3D point spacing implies

an uncertainty in the maximum height being recorded at every local position. This applies in

particular to grass covered surfaces, where individual blades are not necessarily hit by the laser

beam at their highest point. Both of these effects can result in reduced vegetation heights in a

TLS-based approach, compared e.g., to length measurements of individual sedges in the field.

The modular program system OPALS (version 2.3.0, Pfeifer et al., 2014) was used for the point

cloud analyses of ground surface elevation and relative height above the ground surface.





**Acknowledgements.** Logistical support was provided by the Russian-German Samoylov

Research Base (1998–2012) and the Russian Samoylov Island Research Station (2013–2017).

Field support, including data collection, was provided by Konstanze Piel, Steffen Frey, Günter

Stoof, and Waldemar Schneider. We gratefully acknowledge the funding received from the

Helmholtz Association's ACROSS (Advanced Remote Sensing – Ground Truth Demo and Test

Facilities) project.



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
