# Peer review of "A 16-year record (2002–2017) of permafrost, active layer, and meteorological conditions at the Samoylov Island Arctic permafrost research site, Lena River Delta, northern Siberia: an opportunity to validate remote sensing data and land surface, snow, and permafrost models"

_Earth System Science Data, 2018_

## Referee Comment (RC1) · R. L. H. Essery (Referee) · 22 Aug 2018

This is a very valuable dataset, considering the lack of well-maintained and quality-controlled long-term records in the Arctic, but it does not quite live up to the claim of utility for validation of land surface models. Gaps in validation data are not a prob-

lem, but such models cannot handle gaps in their driving data. I entirely understand why measurements of solid precipitation are difficult, but the entire lack of such measurements in particular is fatal. The Level 2 meteorological dataset available from PANGAEA combines data from multiple sensors, but gaps are not filled and physically impossible values have not been replaced. If a fully gap filled and despiked Level 3 meteorological dataset is beyond the scope of this paper, the potential uses of the data would be greatly expanded if the corrected reanalysis data used by Chadburn et al. (2017) could also be included in the archive.

Minor comments

line 32: Snow could also be mentioned as being involved in positive feedbacks and links to energy balance.

line 38: It took me a long time to understand why datasets in this paper start in 2002 if observations began in 1998; I think it is because data from 1998 onwards have already been documented in Boike et al. (2013).

line 50: I would use the English word "level" in place of "niveau".

line 103: "starts at the end of May"

line 116: Is the intended meaning here "at a few" or "at only a few"? The emphases is different.

line 119: Replace "thereof" with "of which"

line 186: The air may have been stagnant, but I think that "constant air temperature values" is intended here.

line 219: Errors for snow-covered radiometers could be much more than 10%, but comparison of outgoing and incoming shortwave radiation gives some indication of when this has occurred (flag 8?).

line 239: Date for placement of the metal plate differs by a year between here and the

caption of Figure B4. Why does the snow depth appear to be constant for a long period in 2017?

line 260: That is Figure 4 of Gouttevin et al., not here.

line 378: In the absence of calibration, why is a probe constant different from the recommendation chosen?

line 499: Incomplete sentence

Figure 4 (which might instead be described as Table 4) misses a bar for 3.5 months of HMP155A.

---

## Referee Comment (RC2) · N. Brown (Referee) · 8 Nov 2018

**General Comments**

The authors present a relevant and useful dataset. The publication of soil moisture data alongside the soil temperature fills a gap in the availability of such measurements (with

the exception of the related Spitsbergen data). These measurements are valuable from a permafrost modelling perspective because they provide a secondary dataset against which model outputs can be validated. A few grammatical errors and unclear sections impede the clarity of the text, but these are easily fixed.

Specific Comments

(multiple locations): Dielectric number vs. dielectric constant vs. relative permittivity. I have most frequently seen the second of these used, although I understand it is falling out of fashion in favor of the third. This is the first time I have seen the term dielectric number used. I see that in appendix A you note the equivalence of these terms, but I think it would be useful to clarify this on the first mention of the term.

L233: After you checked the buckets by pouring in a known volume of water, was this spike removed from the dataset? Were there any instances where the instrument was found to be in error based on this check?

L297: The month, or time of year, would be helpful to contextualize the thaw depth measurements

L335: You mention that the errors are estimated based on the temperature values during zero-curtain periods. Can you be more specific about how exactly you derive the error estimates from these periods?

L450: You introduce level 0, 1 and 2 data but only discuss levels 1 and 2 in the text. Please consider briefly discussing level 0 data as well (PANGAEA): Data are well organized on PANGAEA, but I notice that columns in the soil level 1 dataset are inconsistent between files which will throw off attempts to combine the yearly files. Although this can be solved relatively easily, such a hurdle may make the dataset less accessible to those with little programming experience. Standardizing the columns and simply having NA values in the years where the borehole data are not present would eliminate this problem.

[Figure]

Technical Corrections

L49: "The depth of zero annual amplitude ... has warmed". Clarify here that it is the temperature at that depth which has warmed, not the depth itself

L50: 'niveau' not needed here

L68: "land surface o for" typo

L103 "thawing period starts the end of may". Should read: starts at

L104 "reaches maximum at". Should read: reaches a maximum at

L186: "stagnant". Doesn't seem like the right word choice. Perhaps constant?

L192:"in parallel with". This is more an opinion, but consider using "alongside" to eliminate any possibility that the reader interprets this as the circuitry meaning of 'in paralell'.

L205: 'aligned towards geographic north'. I see in the documentation for the instrument that this refers to the calibration with respect to azimuth. It is a bit confusing in the text as it suggests that the instrument is always pointing north.

L288: "For the unfrozen...". The clarity of this sentence could be improved

L285,287: "signal period" vs "single period" are these meant to be different terms?

L698,700, 712: Significant figures on regression equations – is this level of precision appropriate?

L291: Consider "Instrument installation at the soil station"

L296: Should this subheading have a numerical prefix? 3.2.2?

L306: "Unfrozen topsoil layer" = thaw depth

L310: "Was probed at least once". Would be clearer to say "contained at least one probe"

L376: "and the signal stabilized". Missing word: signal has stabilized (stylistic)

L388: "grounding system improved". Missing word: system was improved (stylistic)

L446: "located on Spitzbergen permafrost" should be "located on Spitzbergen"

L449: "outliers manually" outliers have been manually (stylistic)

L449: "excluded" Doesn't seem like the right word choice, consider 'ruled out'

L457: "combined with data set" should be "combined with a data set"

L467: "flagged in the dataset (Table 3: Flag 7)"

L471: "differences between the locations". Clarify to "the sensor locations"

L474: be consistent with the use of 'record' vs 'data set' (stylistic)

L487: depth of 20.8m.

L499: incomplete sentence: "this makes this an important"
* * *

---

## Author Comment (AC1) · 13 Dec 2018

Dear Richard Essery,

Thank you very much for your detailed comments which helped to clarify the manuscript. We have been through them in detail and made amends as requested. We

provide a point to point response below to your comments (AC), as well as changes in the manuscript (CM). We also provide a revised manuscript in "tracked changes mode". Please note that page numbers refer to the page numbers in the revised manuscript.

On behalf of the authors,

Julia Boike

Please also note the supplement to this comment:
https://www.earth-syst-sci-data-discuss.net/essd-2018-82/essd-2018-82-AC1-supplement.pdf
* * *
[Figure]

**Supplement:**

Dear R. L. H. Essery,

Thank you very much for your detailed comments which helped to clarify the manuscript. We have been through them in detail and made amends as requested. We provide a point to point response below to your comments (AC), as well as changes in the manuscript (CM). We also provide a revised manuscript in "tracked changes mode". Please note that page numbers refer to the page numbers in the revised manuscript.

On behalf of the authors,

Julia Boike

**A 16-year record (2002–2017) of permafrost, active layer, and meteorological conditions at the Samoylov Island Arctic permafrost research site, Lena River Delta, northern Siberia: an opportunity to validate remote sensing data and land surface, snow, and permafrost models**
by Julia Boike et al.

**R. L. H. Essery (Referee)**

| |
|---|
| **RC**: Referee comment \| **AC**: Author comment \| **CM**: Change in the manuscript |

**RC1.01:** This is a very valuable dataset, considering the lack of well-maintained and quality controlled long-term records in the Arctic, but it does not quite live up to the claim of utility for validation of land surface models. Gaps in validation data are not a problem, but such models cannot handle gaps in their driving data. I entirely understand why measurements of solid precipitation are difficult, but the entire lack of such measurements in particular is fatal. The Level 2 meteorological dataset available from PANGAEA combines data from multiple sensors, but gaps are not filled and physically impossible values have not been replaced. If a fully gap filled and despiked Level 3 meteorological dataset is beyond the scope of this paper, the potential uses of the data would be greatly expanded if the corrected reanalysis data used by Chadburn et al. (2017) could also be included in the archive.

**AC1.01:** Several points are raised which we would like to address the following:

*(1) Solid precipitation measurements*
As we did not have continuous power sources until summer 2015, as well as no permanent research base during the fall-winter-spring periods, our instrumentation depended on low energy consumption and non-supervision during these periods. Because of the extreme weather conditions during winter, we often had sensor loss and data gaps. Solid precipitation measurements, for example, heated precipitation gauges, require high power supply as well as supervision. We are currently planning the installation of a heated precipitation gauge for measuring solid (total) precipitation during winter.

*(2) Gap filling, replacement of physically impossible values*
The aim of this paper was to publish the observational data; any gap-filling will require either additional (non-site specific) data sources or interpolation/extrapolation of the observed values. Either way data will be inserted that is not 'real', and there are choices to make about how to do this, which we would leave in the hands of the data user, depending on the purpose for which they want to use it.

We apply a strict quality analysis which also includes a flagging for "physically impossible values". These data are marked in the dataset using the Flag 4 (physical limits: values outside the physically possible or likely limits) which is discussed in the paper (line 560).

*(3) Corrected reanalysis data by Chadburn et al. (2017)*
Based on this comment, we have now archived the gap filled and quality controlled driving data for the Chadburn et al. 2017 paper in PANGEA (Burke et al., 2018). This data set includes all driving data of the PAGE21 sites that have been used and updated for the Chadburn et al. (2017) publication, including the Samoylov site. Eleanor Burke has now been included as a co-author in this ESSD publication since she contributed largely to this driving data dataset.

As described in the paper and on the PANGAEA webpage: These meteorological driving data were prepared using observations from the sites combined with reanalysis data for the grid cell containing the site. For the period 1901–1979, Water and Global Change forcing data (WFD) were used (Weedon et al., 2011). This has half-degree resolution for the whole globe at 3-hourly time resolution from 1901 to 2001. For the period 1979–2014, WATCH-Forcing-Data-ERA-Interim (WFDEI) was used (Weedon, 2013). For the time periods in which observed data were available, correction factors were generated by calculating monthly biases relative to the WFDEI data. These corrections were then applied to the time series from 1979 to 2014 of the WFDEI data. The WFD before 1979 were then corrected to match these data and the two datasets were joined at 1979 to provide gap-free 3-hourly forcing from 1901 to 2014. Local meteorological station observations were used for all variables except snowfall, which was estimated from the observed snow depth by treating increases in snow depth as snowfall events with an assumed snow density. See Chadburn et al. (2017) for more details.

**CM1.01:** Line 184: The gap-free meteorological dataset that was produced and used in Chadburn et al. (2017) is now available on the PANGAEA database (Burke et al., 2018), making it easy for modellers to begin running the Samoylov site and therefore to make good use of our data.

**RC1.02:** line 32: Snow could also be mentioned as being involved in positive feedbacks and links to energy balance.
**AC1.02:** We agree that snow is a very important component of the cryospheric feed backs and added this in the abstract as suggested.
**CM1.02:** Line 37: Permafrost thaw and carbon release into the atmosphere, as well as snow cover changes, are positive feedback mechanisms that have the potential for climate warming.

**RC1.03:** line 38: It took me a long time to understand why datasets in this paper start in 2002 if observations began in 1998; I think it is because data from 1998 onwards have already been documented in Boike et al. (2013).
**AC1.03:** For clarification, we added a sentence following the sentence above.
**CM1.03:** Line 45: Furthermore, we present a merged dataset of the parameters, which were measured from 1998 onwards.

**RC1.04:** line 50: I would use the English word "level" in place of "niveau".

**AC1.04:** We adopted this suggestion.

**CM1.04:** The depth of zero annual amplitude is at 20.75 m. At this depth, the temperature has increased from -9.1 °C in 2006 to -7.7 °C in 2017.

**RC1.05:** line 103: "starts at the end of May"

**AC1.05:** We adopted this suggestion.

**CM1.05:** The active layer thawing period starts at the end of May and active layer thickness reaches maximum at the end of August/beginning of September.

**RC1.06:** line 116: Is the intended meaning here "at a few" or "at only a few"? The emphases is different.

**AC1.06:** We adopted this suggestion.

**CM1.06:** Line 129: Degradation of ice wedges, as observed throughout the Arctic (Liljedahl et al., 2016), occurs at only a few, localized parts of the research site (Kutzbach, 2006). The recent work by Nitzbon et al. (2018) shows that the spatial variability in the types of ice-wedge polygons observed at this study area can be linked to the spatial variability in the hydrological conditions. Furthermore, wetter hydrological conditions have a destabilizing effect on ice wedges and enhance degradation.

**RC1.07:** line 119: Replace "thereof" with "of which"

**AC1.07**: We adopted the suggestion proposed by the referee.

**CM1.07:** The total mapped area of the polygonal tundra on Samoylov Island (excluding the floodplain) is composed of 58% dry tundra, 17% wet tundra and 25% water surfaces, of which 10% are over-grown water and 15% open water (Muster et al., 2012, Figure 3a).

**RC1.08:** line 186: The air may have been stagnant, but I think that "constant air temperature values" is intended here.

**AC1.08:** We adopted the suggestion proposed by the referee.

**CM1.08:** During extreme cold air temperature periods, for example, between 1 February and 10 March 15, 2013, constant air temperature values were recorded at the sensor's output limit.

**RC1.09:** line 219: Errors for snow-covered radiometers could be much more than 10%, but comparison of outgoing and incoming shortwave radiation gives some indication of when this has occurred (flag 8?).

**AC1.09:** We adopted this suggestion and did apply another quality analysis. We flagged radiation data during those time periods where short wave incoming radiation was lower than shortwave outgoing radiation by 10 Wm$^{-2}$ using Flag 6 (plausibility, values unlikely in comparison with other sensor series or for a given time of the year). Using this analysis, 111 out of 134 839 values (30 June 2009 to 21 July 2017), equalling less than 1% of radiation data were flagged during the winter period. Please note that this quality analysis does not differentiate if the sensors were covered by snow or dirt.

**CM1.09:** Line 247: Our quality analysis also includes flagging the data during those periods where short wave incoming was lower than shortwave outgoing by 10 Wm$^{-2}$ using Flag 6 (plausibility, values unlikely in comparison with other sensor series or for a given time of the year). Between 30 June 2009 to 21 July 2017, less than 1% of the data were flagged.

**RC1.10:** line 239: Date for placement of the metal plate differs by a year between here and the caption of Figure B4. Why does the snow depth appear to be constant for a long period in 2017?

**AC1.10:** The reviewer is correct about the date discrepancy and we have corrected the figure legend B4. In 2017, crusted snow occurred on the top of the snow under the sensor and almost all fresh snow was blown away. This can be seen on the time lapse images of the snow surface that are also provided with the data (see section 3.1.6).

**CM1.10:** Figure B4. Campbell Scientific SR50 snow depth sensor, installed on 24 August 2002. An aluminum plate was installed on the ground surface beneath the sensor beam on 17 July 2015.

**RC1.11:** line 260: That is Figure 4 of Gouttevin et al., not here.

**AC1.11:** Adopted as suggested.

**CM1.11:** We have removed the reference to Figure 4 of Gouttevin et al. from the paper. The reference Gouttevin et al. 2018 has been updated in the reference section.

**RC1.12:** line 378: In the absence of calibration, why is a probe constant different from the recommendation chosen?

**AC1.12:** We custom ordered the Campbell Scientific CS605 TDR probes with a length of 20 cm instead of the regularly sold 30 cm. The probe constant is given only for the 30 cm probe in the CSI manual. Thus we entered the multiplier of "1" for later calibration of the probes.

**CM1.12:** Because no calibration was done, and the TDR probes were custom made to 20 cm, a probe constant (Kp) of 1 was used for BEC waveform retrieval;..

**RC1.13:** line 499: Incomplete sentence

**AC1.13:** Adopted as suggested.

**CM1.13:** This makes this an important dataset for modellers.

**RC1.14:** Figure 4 (which might instead be described as Table 4) misses a bar for 3.5 months of HMP155A.

**AC1.14:** As the Vaisala HMP155A was installed on 17 September 2017 and the dataset ends on 21 September 2017 the bar is very thin and hard to recognize.

**CM1.14:** Line 588: Figure 4: Note that the measuring period for the Vaisala HMP155A only started 17 September, 2017, which is why the bar appears very thin.  Recording of all parameters is still continuing at present.

[revised manuscript text omitted]

---

## Author Comment (AC2) · 13 Dec 2018

Dear Nick Brown,

Thank you very much for your detailed comments which helped to clarify the manuscript. We have been through them in detail and made amends as requested. We

provide a point to point response below to your comments (AC), as well as changes in the manuscript (CM). We also provide a revised manuscript in "tracked changes mode".

On behalf of the authors,

Julia Boike

Please also note the supplement to this comment: https://www.earth-syst-sci-data-discuss.net/essd-2018-82/essd-2018-82-AC2-supplement.pdf

**Supplement:**

Dear N. Brown,

Thank you very much for your detailed comments which helped to clarify the manuscript. We have been through them in detail and made amends as requested. We provide a point to point response below to your comments (AC), as well as changes in the manuscript (CM). We also provide a revised manuscript in "tracked changes mode".

On behalf of the authors,

Julia Boike

**A 16-year record (2002–2017) of permafrost, active layer, and meteorological conditions at the Samoylov Island Arctic permafrost research site, Lena River Delta, northern Siberia: an opportunity to validate remote sensing data and land surface, snow, and permafrost models**
by Julia Boike et al.

**N. Brown (Referee)**

**RC**: Referee comment | **AC**: Author comment | **CM**: Change in the manuscript

**RC2.01:** (multiple locations): Dielectric number vs. dielectric constant vs. relative permittivity. I have most frequently seen the second of these used, although I understand it is falling out of fashion in favor of the third. This is the first time I have seen the term dielectric number used. I see that in appendix A you note the equivalence of these terms, but I think it would be useful to clarify this on the first mention of the term.
**AC2.01:** The dielectric constant, dielectric number or relative permittivity all have the same meaning. Dielectric constant was used in earlier times, but since it is not a "constant", we prefer to use dielectric number.
**CM2.01:** The sensor outputs a single period measurement from which usually the bulk dielectric number is calculated. The dielectric number (also referred to as the relative permittivity or dielectric constant) is then used to calculate the volumetric water content using an empirical polynomial calibration provided by the manufacturer.

**RC2.02:** L233: After you checked the buckets by pouring in a known volume of water, was this spike removed from the dataset? Were there any instances where the instrument was found to be in error based on this check?
**AC2.02:** In general, we find that the rain gauges performed very well, even after long periods of non attendance. This could be due to very little vegetation, litter or terrestrial material collected in the rain gauge. The field calibration was done by pouring a known volume of water into the gauge and recording the tipping. We did not find any errors or deviations to

the original calibrations by the manufacturer. The calibration "spikes" were marked in the data set using Flag 3 (example July 16, 2012). Flag 3 (maintenance) is used for all calibration work that is done by the engineers and visible in the data.

**CM2.02:** Line 267: These calibration data are flagged with Flag 3 (maintenance periods).

**RC2.03:** L297: The month, or time of year, would be helpful to contextualize the thaw depth measurements

**AC2.03:** We adopted the suggestion proposed by the referee.

**CM2.03:** A new measurement station was established in August 2002, with instruments installed in four profiles (Appendices B2 and F).

**RC2.04:** L335: You mention that the errors are estimated based on the temperature values during zero-curtain periods. Can you be more specific about how exactly you derive the error estimates from these periods?

**AC2.04:** Because the temperature sensors were installed in the soil, and thus cannot be retrieved for recalibration or repair, we use the zero curtain effect to evaluate the accuracy of the probes. During the freeze back period, the soil temperature data are around 0 °C due to the phase change of water to ice. Basically, we use this zero curtain effect as ice bath calibration where the accuracy is tested again 0°C of ice slush water. During these phase change periods we obtained the maximum deviation to 0 °C to something up to ±0.7 °C for 2009.

**CM2.04:** Line 375: The zero curtain period during fall – winter, where temperatures in the ground are stabilized at 0°C during phase change, offers an accuracy test for sensors that cannot be retrieved.

**RC2.05:** L450: You introduce level 0, 1 and 2 data but only discuss levels 1 and 2 in the text. Please consider briefly discussing level 0 data as well (PANGAEA): Data are well organized on PANGAEA, but I notice that columns in the soil level 1 dataset are inconsistent between files which will throw off attempts to combine the yearly files. Although this can be solved relatively easily, such a hurdle may make the dataset less accessible to those with little programming experience. Standardizing the columns and simply having NA values in the years where the borehole data are not present would eliminate this problem.

**AC2.05:** We added NA values for the borehole in the years 2002-2005 to make the dataset consistent and more easily accessible. We also performed a second quality check of all data sets again. These revised data are now available on PANGAEA and Zenodo.

We also added information on Level 0 in the text as well as in Table 3.

**CM2.05**: Line 493: Level 0 are data with equal time steps (UTC), data gaps filled with NA and standardized into one file format. These data, as well as raw data, are stored internally at AWI and are not archived in PANGAEA.

Page 38, line 580: (Table 3): Standardized format with data in equal time steps (UTC), filled with NA for data gaps.

**RC2.06:** L49: "The depth of zero annual amplitude . . . has warmed". Clarify here that it is the temperature at that depth which has warmed, not the depth itself

**AC2.06**: Adopted as suggested.

**CM2.06:** The depth of zero annual amplitude level is at 20.75 m. At this depth, the temperature has increased from -9.1 °C in 2006 to -7.7 °C in 2017.

**RC2.07:** L50: 'niveau' not needed here

**AC2.07:** We adopted the suggestion proposed by the referee.

**CM2.07:** The depth of zero annual amplitude is at 20.8 m, and has warmed from -9.1 °C in 2006 to -7.7 °C in 2017.

**RC2.08:** L68: "land surface o for" typo

**AC2.08:** We adopted the suggestion proposed by the referee.

**CM2.08:** The seasonal snow cover in Arctic permafrost regions can blanket the land surface  for many months of the year and has an important effect on the thermal regime of permafrost-affected soils (Langer et al., 2013).

**RC2.09:** L103 "thawing period starts the end of may". Should read: starts at

**AC2.09:** We adopted the suggestion proposed by the referee.

**CM2.09:** The active layer thawing period starts at the end of May and active layer thickness reaches maximum at the end of August/beginning of September.

**RC2.10:** L104 "reaches maximum at". Should read: reaches a maximum at

**AC2.10:** We adopted the suggestion proposed by the referee.

**CM2.10:** The active layer thawing period starts at the end of May and active layer thickness reaches a maximum at the end of August/beginning of September.

**RC2.11:** L186: "stagnant". Doesn't seem like the right word choice. Perhaps constant?

**AC2.11:** We adopted the suggestion proposed by the referee.

**CM2.11:** During extreme cold air temperature periods, e.g., between 1 February and 15 March 2013, constant air temperature values were recorded at the sensor's output limit.

**RC2.12:** L192:"in parallel with". This is more an opinion, but consider using "alongside" to eliminate any possibility that the reader interprets this as the circuitry meaning of 'in paralell'.

**AC2.12:** We adopted the suggestion proposed by the referee.

**CM2.12:** Campbell Scientific PT100 temperature sensors were installed on 22 August 2013 alongside the temperature and humidity probes, at the same heights but in separate unventilated shields, in order to circumvent this problem.

**RC2.13:** L205: 'aligned towards geographic north'. I see in the documentation for the instrument that this refers to the calibration with respect to azimuth. It is a bit confusing in the text as it suggests that the instrument is always pointing north.

**AC2.13:** Corrected as suggested.

**CM2.13:** This was done by orienting the center line of the sensor towards true north (using a GPS reference point) and then rotating the sensor base until the datalogger indicated zero degrees.

**RC2.14:** L288: "For the unfrozen. . .". The clarity of this sentence could be improved

**AC2.14**: For the unfrozen periods, the soil as measured by a dielectric device is a mixture of air, water, and soil particles.

**CM2.14:** For the unfrozen periods, the soil as measured by a dielectric device is a mixture of air, water, and soil particles.

**RC2.15:** L285,287: "signal period" vs "single period" are these meant to be different terms?

**AC2.15:** We corrected the typo, the correct wording is "signal period".

**CM2.15:** The sensor outputs a signal period measurement from which usually the bulk dielectric number is calculated.

**RC2.16:** L698,700, 712: Significant figures on regression equations – is this level of precision appropriate?

**AC2.16:** Note that the high precision of the fitting parameters is required, since the WL (in cm) estimated by equations (D1) and (D2) is very sensitive to these parameters.

**CM2.16:** No changes in the text.

**RC2.17:** L291: Consider "Instrument installation at the soil station"

**AC2.17:** We adopted the suggestion proposed by the referee.

**CM2.17:** 3.2.1 Instrument installation at the soil station and soil sampling

**RC2.18:** L296: Should this subheading have a numerical prefix? 3.2.2?

**AC2.18:** This subheading is hierarchically under the 3.2.1 heading. As ESSD does not want fourth level headings, this heading has not a numerical prefix.

**CM2.18:** No change in the manuscript.

**RC2.19:** L306: "Unfrozen topsoil layer" = thaw depth

**AC2.19:** Adopted as suggested.

**CM2.19**: The thaw depth was between 17 and 40 cm thick at the time of instrument installation.

**RC2.20:** L310: "Was probed at least once". Would be clearer to say "contained at least one probe"

**AC2.20:** We adopted the suggestion proposed by the referee.

**CM2.20:** The sensors were positioned according to the soil horizons so that every horizon in the profile contained at least one probe.

**RC2.21:** L376: "and the signal stabilized". Missing word: signal has stabilized (stylistic)

**AC2.21:** We adopted the suggestion proposed by the referee.

**CM2.21:** The impedance can be determined from the attenuation of the electromagnetic wave traveling along the TDR probe after all multiple reflections have ceased and the signal has stabilized.

**RC2.22:** L388: "grounding system improved". Missing word: system was improved (stylistic)

**AC2.22:** We adopted the suggestion proposed by the referee.

**CM2.22:** Data quality improved significantly after August 2015 when the Campbell Scientific coaxial SDMX50 multiplexers were exchanged for SDM8X50 and the electrical grounding system was improved.

**RC2.23:** L446: "located on Spitzbergen permafrost" should be "located on Spitzbergen"

**AC2.23:** Adopted as suggested.

**CM2.23:** located on Spitzbergen "

**RC2.24:** L449: "outliers manually" outliers have been manually (stylistic)
**AC2.24:** We adopted the suggestion proposed by the referee.
**CM2.24:** In addition to the automated processing, all data have been visually controlled and outliers have been manually detected, but it cannot be excluded that there are still unreasonable values present which are not flagged accordingly.

**RC2.25:** L449: "excluded" Doesn't seem like the right word choice, consider 'ruled out'
**AC2.25:** We adopted the suggestion proposed by the referee.
**CM2.25:** In addition to the automated processing, all data have been visually controlled and outliers have been manually detected, but it cannot be ruled out that there are still unreasonable values present which are not flagged accordingly.

**RC2.26:** L457: "combined with data set" should be "combined with a data set"
**AC2.26:** We adopted the suggestion proposed by the referee.
**CM2.26:** Examples in this paper of Level 2 data are soil temperature and meteorological data (air temperature, humidity, wind speed, and net radiation) recorded between 1998–2002 (Boike et al., 2013) that have been combined with a data set since 2002 into a single data series, in order to obtain a long term picture (documentation of source data is provided in the PANGAEA data archives).

**RC2.27:** L467: "flagged in the dataset (Table 3: Flag 7)"
**AC2.27:** We adopted the suggestion proposed by the referee.
**CM2.27:** Our temperature data have been checked against the fall zero-curtain effect and information on any reduction in accuracy is flagged in the data set (Flag 7: decreased accuracy; Table 3).

**RC2.28:** L471: "differences between the locations". Clarify to "the sensor locations"
**AC2.28:** We adopted the suggestion proposed by the referee.
**CM2.28:** The local differences between the sensor locations from 1998 and 2002 (even though less than 50 m meters apart), as well as differences between sensor types and accuracies, need to be considered when interpreting longer term records.

**RC2.29:** L474: be consistent with the use of 'record' vs 'data set' (stylistic)
**AC2.29:** We adopted the suggestion proposed by the referee.
**CM2.29:** E.g., relative air humidity data show marked differences between the earlier data set (1998–1999) compared to the later data set (starting in 2002).

**RC2.30:** L487: depth of 20.8m.
**AC2.30:** We adopted the suggestion proposed by the referee. We also changed the depth of 20.8 to 20.75 m (as in Pangea) since this is the exact installation depth.
**CM2.30:** Since the installation in 2006, permafrost has warmed by 1.3 °C at the zero annual amplitude depth at 20.75 m .

**RC2.31:** L499: incomplete sentence: "this makes this an important"
**AC2.31:** Adopted as suggested.

[revised manuscript text omitted]

---

## Author Response (AR2)

Dear Kirsten,

Thank you very much for your comments. We provide a point to point response below to your comments (EC), as well as changes in the manuscript. We also provide a revised manuscript in "tracked changes mode".
On behalf of the authors,

Julia Boike

**A 16-year record (2002–2017) of permafrost, active layer, and meteorological conditions at the Samoylov Island Arctic permafrost research site, Lena River Delta, northern Siberia: an opportunity to validate remote sensing data and land surface, snow, and permafrost models**
by Julia Boike et al.

**Kirsten Elger (Editor)**

**EC 1**. Please make sure the DOI of Boike et al. (2018) is registered with PANGAEA (https://doi.org/10.1594/PANGAEA.891142, the DOI is reloving with the pangaea DOI resolver, but not via doi.org, this is also the case for Burke et al, 2018)

**Answer**: The DOI is now registered.

**EC2**. Please cite the Boike et al. dataset in the manuscript (e.g. in the data description and data availability sections) and add the following reference to the reference list: Boike, J., Nitzbon, J., Anders, K., Grigoriev, M. N., Bolshiyanov, D. Yu, Langer, M., Lange, S., Bornemann, N., Morgenstern, A., Schreiber, P., Wille, C., Chadburn, S., Gouttevin, I., Kutzbach, L.: Measurements in soil and air at Samoylov Station (2002-2018). PANGAEA, https://doi.pangaea.de/10.1594/PANGAEA.891142. 2018.

**Answer**: The reference to the dataset has been updated in the data description (Boike et al. 2018, a, b,c) and reference list:

Boike, J., Nitzbon, J., Anders, K., Grigoriev, M. N., Bolshiyanov, D. Yu, Langer, M., Lange, S., Bornemann, N., Morgenstern, A., Schreiber, P., Wille, C., Chadburn, S., Gouttevin, I., Kutzbach, L.: Measurements in soil and air at Samoylov Station (2002-2018). PANGAEA, https://doi.pangaea.de/10.1594/PANGAEA.891142, 2018a.

Boike, J., Nitzbon, J., Anders, K., Grigoriev, M.N., Bolshiyanov, D.Y., Langer, M., Lange, S., Bornemann, N., Morgenstern, A., Schreiber, P., Wille, C., Chadburn, S., Gouttevin, I. and Kutzbach, L.: TLS measurements at Samoylov in 2017, link to pointsclouds and DEM, PANGAEA, https://doi.pangaea.de/10.1594/PANGAEA.891157, 2018b.

Boike, J., Nitzbon, J., Anders, K., Grigoriev, M.N., Bolshiyanov, D.Y., Langer, M., Lange, S., Bornemann, N., Morgenstern, A., Schreiber, P., Wille, C., Chadburn, S., Gouttevin, I. and Kutzbach, L.. Time

lapse camera pictures at Samoylov, LTO, 2002-2017, PANGAEA., https://doi.pangaea.de/10.1594/PANGAEA.891129, 2018c

**EC3**. Please make sure the DOI of Burke et al (2018) is registered with PANGAEA (https://doi.org/10.1594/PANGAEA.896133) and remove the "(DOI registration in progress)" from the reference list

**Answer**:
Burke et al. (2018) is now registered (https://doi.pangaea.de/10.1594/PANGAEA.896133) and corrected in the reference list.

**EC suggestion**: it would be very helpful explain the content of the three Zenodo-DOIs in the data availability statement. The Pangaea DOI is a collection with seven datasets, which makes it difficult to translate the three Zenodo-DOis to the Pangaea version. I also wonder if the data in Zenodo and Pangaea are the same? If yes, it would me much clearer to the reader to only privide the link to the collection in Pangaea. Formally, the Zenodo-DOIs should also be cited in the reference list. My suggestion is to delete the reference to Zenodo and make sure the latest version of the data is published with PANGAEA.

**Answer:**

The PANGAEA and Zenodo archives include the same dataset (share the same DOIs) and are now identically organized with the following structure:

PANGAEA
Time series (https://doi.pangaea.de/10.1594/PANGAEA.891142)
Terrestrial laser scanning (https://doi.pangaea.de/10.1594/PANGAEA.891157)
Time lapse camera images (https://doi.pangaea.de/10.1594/PANGAEA.891129)

Zenodo:
Time series (https://zenodo.org/record/2223709)
Terrestrial laser scanning (https://zenodo.org/record/2222569)
Time lapse camera images (https://zenodo.org/record/2222454)

We have rewritten the data availability section:

*The data sets presented herein can be downloaded from PANGAEA (https://www.pangaea.de/) and Zenodo (https://zenodo.org/), which provides dataset view and download statistics. Data (including links to subsets) can be found on either repository using the following links: https://doi.pangaea.de/10.1594/PANGAEA.891142 and https://zenodo.org/record/2223709 . Permafrost temperature and active layer thaw depth data are also available through the Global Terrestrial Network for Permafrost (GTN-P) database (http://gtnpdatabase.org).*

We would also like to include some further information why we have archived the data in two archives, also following a recent discussion with David Carlson (December 2018).

In our opinion, both archives offer different pros and contras, which we will not detail here. One important difference and very important is, that we need to provide statistics for EU reporting as part of INTERACT virtual Access for Samoylov. This statistics on how often datasets are downloaded and viewed  is only provided through Zenodo.

Please note that we also contribute the data to additional permafrost databases, such as GTNP and CALM which is the preferred archive for permafrost temperatures and thaw depth.

[revised manuscript text omitted]